# Target Enclosing and Coverage Control for Quadrotors with Constraints and Time-Varying Delays: A Neural Adaptive Fault-Tolerant Formation Control Approach

**DOI:** 10.3390/s22197497

**Published:** 2022-10-02

**Authors:** Ziqian Zhao, Ming Zhu, Xiaojun Zhang

**Affiliations:** 1School of Aeronautic Science and Engineering, Beihang University, Beijing 100191, China; 2Institute of Unmanned System, Beihang University, Beijing 100191, China

**Keywords:** time-varying formation, time-delay, RBFNN, fault-tolerant control, adaptive control, state constraint

## Abstract

This paper investigates the problem of formation fault-tolerant control of multiple quadrotors (QRs) for a mobile sensing oriented application. The QRs subject to faults, input saturation and time-varying delays can be controlled to perform a target-enclosing and covering task while guaranteeing the state constraints will not be exceeded. A distributed formation control scheme is proposed, using a radial basis function neural network (RBFNN)-based time-delay position controller and an adaptive fault-tolerant attitude controller. The Lyapunov–Krasovskii approach is used to analyze the time-varying delay. Barrier Lyapunov function is deployed to handle the prescribed constraints, and an auxiliary system combined with a command filter is designed to resolve the saturation problem. An RBFNN and adaptive estimators are deployed to provide estimates of disturbances, fault signals and uncertainties. It is proven that all the closed-loop signals are bounded under the proposed protocol, while the prescribed constraints will not be violated, which enhances the flight safety and QR formation’s applicability. Comparative simulations based on application scenarios further verify the effectiveness of the proposed method.

## 1. Introduction

Formation control technology, which is based on the theory of multi-agent systems (MAS), enables multiple unmanned aerial vehicles (UAVs) to efficiently complete a shared task and is widely used in aerial mapping, atmospheric environment monitoring and even coordinated military missions [1,2,3].

As a typical small-scale UAV, quadrotor (QR) is qualified to be a formation platform for a variety of applications due to its simple structure, strong maneuverability and hovering capability [4], particularly for mobile sensing tasks, such as target-enclosing and covering, which have been studied by several works so far. The main purpose of the former was to control several mobile sensors to rotate around or above a detected target to obtain detailed information from all angles [5,6,7]. The objective of the latter was to optimize the deployment location of multiple sensors to achieve effective coverage of the interest area, where the methods are mainly Voronoi partitioning-based [8,9], coverage cost function-based [10], K-means-based [11] and reinforcement learning-based [12]. However, these methods cannot be directly applied to small-scale aerial platforms due to the contradiction between the complex location optimization algorithms and limited computing resources. In this paper, a consensus-based formation controller was designed. The UAV’s movement and placement can be directly and flexibly set by time-varying formation functions and virtual leader trajectory, which ensures that the mobile sensing task, including the above two, can be performed when the formation tracking is realized by QR members.

With the expansion of formation technology applications, formation operation reliability has become more prominent, with the fault-tolerant control (FTC) being one of the most important factors. Due to the possibility of a topological chain reaction [13,14], a formation composed of multiple interconnected individuals is more susceptible to malfunction effects than a single system. In recent years, formation FTC has garnered considerable interest, and typical methods can be categorized as either active or passive. In the active FTC design, actuator faults are diagnosed, and parameters can be reconfigured online to achieve the desired performance [15]. An iterative learning observer-based reconstructive-FTC protocol for spacecraft formation was designed in [16]. A reinforcement learning-based data-driven active FTC method for multiple QRs was studied in [17]. However, active FTC approaches are also difficult to implement with small UAVs due to their complexity and high computational requirements. In contrast, the passive FTC requires less computational power due to its algorithm’s simplicity [18,19]. The actuator fault effect for multiple aircraft is addressed in [20] using an adaptive $H_{\infty }$ scheme. A projection-based adaptive FTC protocol was proposed in [21] for a group of UAV formation systems. In [22], a sliding mode-based adaptive FTC scheme for a heterogeneous MAS is presented. Taking into account the limited computing resources of QRs, the adaptive FTC method is adopted in this paper, which is one of the state-of-the-art FTC methods.

Even though UAV formation FTC has made significant progress, there are still problems, obstacles and limitations to its practical application. The fact that practical engineering systems have limitations is one of them. On the one hand, due to the hardware’s physical limitations, the control forces and torques generated by UAV actuators are naturally constrained, also known as the saturation phenomenon, which may result in a decline in performance [23]. In [24], anti-windup compensators are employed against input saturations of a linear MAS. In [25], an auxiliary dynamic system is introduced to address the saturation problem for multiple UAVs. On the other hand, due to safe operation or system-specific requirements, certain UAV system states must be constrained. For example, Some sensor payloads that directly attached to the UAV frame require that the UAV’s attitude angular velocity be constrained within the sensor’s allowed range, and the pan-tilt-zoom (PTZ) system used to stabilize optical sensors also has constraint requirements on UAV’s attitude states [26]. Such consideration is crucial, particularly when actuator faults exist and may result in constraints being violated. According to [27], the associated state constraint problem for a second-order MAS was resolved using a combination of the barrier function and sliding mode control technique. According to [28], motion and visibility constraint problems for multiple robots were resolved by planning a feasible trajectory. By employing performance function and error transformation, Ref. [29] solved the field of view constraint problem for mobile robots formation. However, without modifying the control structure, the methods in [27,28,29] cannot be applied to unconstrained scenarios. Moreover, the aforementioned two types of constraints are typically studied separately and have never been investigated simultaneously in the formation FTC domain.

In addition, due to the formation network’s limited communication capabilities, time delays are unavoidable, which may reduce system performance [30,31]. Based on LMIs theory, Ref. [32] solved the equality communication time delay problem for a group of UAVs. By developing the Lyapunov–Krasovskii (L–K) function, Ref. [33] addressed time-varying delay problem for a 2nd-order MAS. By applying generalized Halanay inequality, Ref. [34] investigated the formation tracking control of 2nd-order MAS with time-varying delays. However, the formation configuration cannot be adjusted dynamically in these works, limiting the application scope. In addition, wind disturbance has a significant impact on the movement of small UAVs in the real world, particularly when the modeling is inaccurate. To circumvent this issue, the mainstream techniques typically include neural networks estimators [35,36,37], nonlinear observers [38,39] and adaptive estimators [40,41], etc. On the basis of the aforementioned factors, we neutralize the effect of disturbances, uncertainties and time-varying communication delays and achieve precise control of time-varying formations.

In light of the aforementioned obstacles, we propose a novel QR formation FTC framework for a mobile sensing oriented application. The main contribution of this work is threefold. Firstly, based on a distributed adaptive FTC mechanism, the effect of time-varying multiplicative and additive faults can be effectively compensated for each QR, and the desired formation flight can still be achieved. Secondly, by applying the barrier Lyapunov function (BLF) technique and designing an auxiliary system, the attitude states of QRs can be constrained in the presence of input saturation, and our BLF analysis can also be applied to unconstrained scenarios without modifying the control structure. Compared to the methods in [27,28,29], the scope of application is expanded. Thirdly, the time-varying delay of each QR is different. Only delayed neighbor information is needed to realize formation flight; that is, the proposed protocol is distributed, and the time-varying formation configuration can be flexibly designed to adapt to target enclosing, area covering and other scenarios. Meanwhile, the disturbances and uncertainties are handled properly by radial basis function neural network (RBFNN) and an adaptive estimator; the application restrictions in real-word environments are relaxed compared to [32,33,34].

Notations: Let a⊗b denote the Kronecker product of matrices a and b, and $\sigma_{min}\left(•\right)$ and $\sigma_{max}\left(•\right)$ indicate the minimum and maximum singular value of a matrix. We denote |•| as the absolute value of a real number, ∥•∥ the Euclidean norm of a vector and ${∥•∥}_{F}$ the Frobenius norm of a matrix.

## 2. Preliminaries and Problem Formulation

### 2.1. Basic Concepts on Graph Theory

An undirected graph G={V,E,W} represents the communication topology of the *N* QRs, which contains a set of nodes $V=\{q_{1},q_{2},...,q_{N}\}$, a set of edges $E\subseteq \{\left(q_{i},q_{j}\right):q_{i},q_{j}\in V\}$ and a weighted adjacency matrix $W=\left[a_{ij}\right]\in R^{N\times N}$. If agent *i* is connected by an edge with agent *j*, that is $(q_{i},q_{j})\in E$, then $a_{ij}=a_{ji}>0$. Otherwise $a_{ij}=a_{ji}=0$, and $a_{ii}=0$ for all i∈Σ={1,2,...,N}. The set of neighbors of node $q_{i}$ is defined by $N_{i}=\{q_{j}\in V:\left(q_{i},q_{j}\right)\in E\}$. The out-degree of node $q_{i}$ is defined by $Deg_{out}\left(q_{i}\right)=\begin{array}{c} \sum_{j\in N_{i}}a_{ij} \end{array}$. The degree matrix of graph G is represented by $D=diag\{Deg_{out}\left(q_{i}\right),i\in \Sigma \}$, and the Laplacian matrix of graph G is represented by L=D−W. The undirected graph G is said to be connected if a path exists between any two nodes $q_{i},q_{j}\in V$, where the path represents a series of diverse adjacent points from $q_{i}$ to $q_{j}$. If $q_{i}$ can access information from the leader, then the connection weight between them $b_{i}>0$. Otherwise $b_{i}=0$, and the matrix form is $B=diag\{b_{1},b_{2},...,b_{N}\}$. Throughout this brief, the following assumption is made for the communication topology.

**Assumption** **1.**
*The undirected graph for N QRs is connected, and there exists at least one path between the leader and follower.*


### 2.2. Problem Formulation and Modeling

Consider a group of *N* QRs following a virtual leader labeled as 0, of which the interaction topology is described by an undirected graph G; it is assumed that graph G is connected. Taking practical factors into account, the dynamic model of QR can be formulated by using Newton’s laws [42]:(1)$$\left\{ \begin{array}{c} {\dot{P}}_{i}=V_{i}\\ {\dot{V}}_{i}=-ge_{3}+R_{it}\frac{T_{i}}{m_{i}}e_{3}+F_{iP}\left(P_{i},V_{i},t\right) \end{array}\;\;\right.$$
(2)$$\left\{ \begin{array}{c} {\dot{A}}_{i}=R_{ir}\Omega_{i}\\ {\dot{\Omega }}_{i}=\Delta F_{i}F_{iA}+J_{i}^{-1}U_{iA}+D_{iA} \end{array}\;\;\right.$$
where i∈Σ, $P_{i}={\left[P_{i1},P_{i2},P_{i3}\right]}^{T}$, $V_{i}={\left[V_{i1},V_{i2},V_{i3}\right]}^{T}$ and $A_{i}={\left[\phi_{i},\theta_{i},\psi_{i}\right]}^{T}$ are position, velocity and attitude of the *i*-th QR in inertia frame, respectively. $\Omega_{i}={\left[p_{i},q_{i},r_{i}\right]}^{T}$ represents the angular velocity in a body-fixed frame. In addition, $m_{i}$ and $J_{i}=diag\{J_{xi},J_{yi},J_{zi}\}$ represents the total mass and inertia matrix, respectively. $T_{i}$ represents the total thrust, $e_{3}={\left[0,0,1\right]}^{T}$ and g represents the gravity constant, and $F_{iP}\left(P_{i},V_{i},t\right)={\left[f_{i1},f_{i2},f_{i3}\right]}^{T}$ represents the lumped uncertainty term including disturbance and inaccurate modeling in (1). $F_{iA}=-J_{i}^{-1}S\left(\Omega_{i}\right)J_{i}\Omega_{i}-J_{i}^{-1}K_{iD}\Omega_{i}$, $K_{iD}=diag\{K_{i1D},K_{i2D},K_{i3D}\}$ represents the aerodynamic damping coefficient. $U_{iA}={\left[U_{i1A},U_{i2A},U_{i3A}\right]}^{T}$ represent the control input torque. Unknown time-varying function $\Delta F_{i}\in R^{3\times 3}$ and $D_{iA}$ represent parameter perturbation and external disturbance in (2), respectively. $R_{it}$, $R_{ir}$ and $S(\Omega_{i})$ are shown below: $$\begin{array}{c} R_{ti}=[\begin{array}{ccc} c_{\theta_{i}}c_{\psi_{i}} & s_{\theta_{i}}c_{\psi_{i}}s_{\phi_{i}}-s_{\psi_{i}}c_{\phi_{i}} & s_{\theta_{i}}c_{\psi_{i}}c_{\phi_{i}}+s_{\psi_{i}}s_{\phi_{i}}\\ c_{\theta_{i}}s_{\psi_{i}} & s_{\theta_{i}}s_{\psi_{i}}s_{\phi_{i}}+c_{\psi_{i}}c_{\phi_{1}} & s_{\theta_{i}}s_{\psi_{i}}c_{\phi_{i}}-c_{\psi_{i}}s_{\phi }\\ -s_{\theta_{i}} & c_{\theta_{i}}s_{\phi_{i}} & c_{\theta_{i}}c_{\phi_{i}} \end{array}],\; \end{array}$$
$$R_{ri}=\frac{1}{c_{\theta_{i}}}[\begin{array}{ccc} c_{\theta_{i}} & s_{\phi_{i}}s_{\theta_{i}} & c_{\phi_{i}}s_{\theta_{i}}\\ 0 & c_{\phi_{i}}c_{\theta_{i}} & -s_{\phi_{i}}c_{\theta_{i}}\\ 0 & s_{\phi_{i}} & c_{\phi_{i}} \end{array}],\;S\left(\Omega_{i}\right)=[\begin{array}{ccc} 0 & -r_{i} & q_{i}\\ r_{i} & 0 & -p_{i}\\ -q_{i} & p_{i} & 0 \end{array}],\;$$
where $s_{\left(*\right)}≜\sin\left(*\right),c_{\left(*\right)}≜\cos\left(*\right)$.

The model of input saturation is expressed as follows:(3)$$U_{ikA}=\{\begin{array}{cc} U_{ik,H}, & if\;U_{ikAF}>U_{ik,H}\\ U_{ikAF}, & if\;U_{ik,L}\leq U_{ikAF}\leq U_{ik,H}\\ U_{ik,L}, & if\;U_{ikAF}<U_{ik,L} \end{array}\;\;$$
where k=1,2,3, $U_{iAF}\left(t\right)={\left[U_{i1AF},U_{i2AF},U_{i3AF}\right]}^{T}\in R^{3}$ represents the control input free from limits but subject to actuator faults, which are expressed as follows
(4)$$U_{iAF}\left(t\right)=\Gamma_{i}\left(t\right)U_{iAC}\left(t\right)+\delta_{i}\left(t\right)\;\;$$
where $\delta_{i}\left(t\right)={\left[\delta_{i1},\delta_{i2},\delta_{i3}\right]}^{T}\in R^{3}$ and $\Gamma_{i}\left(t\right)=diag\{\Gamma_{i1},\Gamma_{i2},\Gamma_{i3}\}\in R^{3\times 3}$ are time-varying additive and multiplicative actuator faults, respectively. $U_{iAC}\left(t\right)\in R^{3}$ is generated by the attitude controller to be designed.

Considering that most sensors and PTZ systems have constraint requirements for rotational motion, the attitude states of QRs will be constrained and are defined as follows
(5)$$\left\{ \begin{array}{c} ∥A_{i}∥<{\bar{C}}_{i1}\left(t\right)\\ ∥\Omega_{i}∥<{\bar{C}}_{i2}\left(t\right) \end{array}\;\;\right.$$
where ${\bar{C}}_{im}\left(t\right)\in R,m=1,2$ represents the time-varying constraints.

The formation center is regarded as the virtual leader, which is specified by $P_{0}=\frac{1}{N}\begin{array}{c} \sum_{i}^{N}P_{i} \end{array}$, and its trajectory is $P_{d}={\left[P_{1d}\left(t\right),P_{2d}\left(t\right),P_{3d}\left(t\right)\right]}^{T}$, which is piecewise 2nd-order differentiable. The time-varying formation pattern is set by a vector $\Lambda_{F}\left(t\right)={\left[\Lambda_{1}^{T}\left(t\right),\Lambda_{2}^{T}\left(t\right),...,\Lambda_{N}^{T}\left(t\right)\right]}^{T}$ with the geometric center set as $\Lambda_{0}=\frac{1}{N}\begin{array}{c} \sum_{1}^{N}\Lambda_{i}\left(t\right) \end{array}$, where $\Lambda_{i}={\left[\Lambda_{i1}\left(t\right),\Lambda_{i2}\left(t\right),\Lambda_{i3}\left(t\right)\right]}^{T},$i=1,⋯,N, $\Lambda_{ik}\left(t\right)$ are 2nd-order differentiable functions defining the motion mode of *i*-th QR with respect to the geometric center, k=1,2,3. Based on consensus theory, we give the following definition:

**Definition** **1.**
*The formation tracking flight is said to be achieved when*

(6)
$$\left\{ \begin{array}{c} \lim_{t\to \infty }\left(P_{i}-P_{j}\right)=\Delta_{ij},\forall i,j\in \Sigma \\ \lim_{t\to \infty }P_{0}=P_{d} \end{array}\;\;\right.$$

*where $\Delta_{ij}=\Lambda_{i}-\Lambda_{j}$.*


Except the communication delay $\tau_{ij}\left(t\right)$ between *i*-th QR and *j*-th QR, this paper also considers the self delay $\tau_{ii}\left(t\right)$ of *i*-th QR caused by calculation or measurement. $\tau_{ij}\left(t\right)$ and $\tau_{ii}\left(t\right)$ are generally regarded as uniform delay $\tau_{i}\left(t\right)$ in the MAS consensus control problem [43].

**Assumption** **2.**
*The time-varying delay has upper bound, that is, $\tau_{i}\leq \tau_{M}$, i∈Σ.*


### 2.3. Control Objective

As depicted in Figure 1, the objective of this work is to design a formation control scheme for the QR mobile sensing platforms to perform the following tasks. The first one is a covering task, in which the QRs can follow the virtual leader to track a moving target and fully cover the target’s adjacent area to carry out sensing or surveillance. The second task is target enclosing, in which the QRs can be controlled to gather and rotate above the moving target to monitor or observe it. The detailed control objectives of proposed formation control protocol are as follows:Consensus-based time-varying formation control protocol (10) is designed based on the demands of the target-enclosing and covering tasks;Distributed adaptive FTC mechanism is deployed to compensate the fault signals (4);BLF and auxiliary system are designed to ensure that the constraint requirements (5) of sensor payload will not be violated in the presence of input saturation (3);The influence of time-varying communication delay $\tau_{i}$ can be eliminated by the L–K technique;The problem of uncertainties and disturbances in (1) and (2) can be neutralized RBFNN (12) and adaptive estimators (49).

## 3. Main Results

The desired formation control scheme is proposed in Figure 2, which can be divided into a RBFNN-based time-delay position controller (NTDPC) (outer-loop) and an adaptive fault-tolerant attitude controller (AFTAC) (inner-loop). The inputs of outer-loop, including time-delayed neighbor information ${\left(P_{j\tau },V_{j\tau }\right)}_{j\in N_{i}}$, time-delayed self information $P_{i\tau },V_{i\tau }$ and time-delayed leader information $P_{d\tau },\Lambda_{F\tau }$ are entered to NTDPC. In the mean time, the lumped uncertainties $F_{iP}\left(P_{i},V_{i},t\right)$ are compensated by the RBFNN approximation law. Then, the command attitude signals $\phi_{iC},\theta_{iC}$ and total thrust $T_{iC}$ are calculated from the outputs of NTDPC. The inputs of inner-loop, including command attitude signals $\phi_{iC},\theta_{iC},\psi_{iC}$, are transferred to AFTAC. Meanwhile, the external disturbances $D_{iA}$, actuator faults $\Gamma_{i},\delta_{i}$ and model uncertainties $\Delta F_{i}$ are compensated by adaptive estimation laws. Finally, the control inputs $U_{iA}$ and $T_{id}$ are applied to *i*-th QR for formation flight. It should be pointed out that the derivatives of $\phi_{iC},\theta_{iC}$ are obtained from Command Filter_1 for the sake of reducing computational burden.

### 3.1. RBFNN Approximation

Suppose an unknown smooth nonlinear function $f\left(x\right):R^{m}\to R$ can be approximated over a prescribed compact set $\Sigma_{R}\in R^{m}$ as follows
(7)$$f\left(x\right)=W^{*T}\Psi \left(x\right)+\epsilon \;\;$$
where $\Psi \left(x\right)={\left[\psi_{1}\left(x\right),\cdots ,\psi_{l}\left(x\right)\right]}^{T}:\Sigma_{R}\to R^{l}$ denotes the radial basis function vector, of which the element is expressed as follows
$$\psi_{k}\left(x\right)=exp\left(-\frac{{\left(∥x-\varsigma_{k}∥\right)}^{2}}{\mu_{k}^{2}}\right),k=1,\cdots ,l$$
where $\varsigma_{k}\in R^{m}$ and $\mu_{k}\in R$ are the center and spread. ϵ∈R is the bounded RBFNN approximation error on $\Sigma_{R}$, that is, $|\epsilon |\leq \bar{\epsilon }$ with $\bar{\epsilon }$ is an unknown constant. $W^{*}\in R^{l}$ is the ideal RBFNN weight vector expressed as follows
$$W^{*}=arg\min_{\hat{W}}\{\underset{x\in \Sigma_{R}}{\sup}|f\left(x\right)-{\hat{W}}^{T}\Psi \left(x\right)|\}$$
where $\hat{W}$ represents the estimation of $W^{*}$.

### 3.2. Design of NTDPC

For *i*-th QR, the local tracking errors are defined as follows
(8)$$e_{iP}=\sum_{j\in N_{i}}a_{ij}\left[P_{i}\left(t\right)-P_{j}\left(t\right)-\Delta_{ij}\left(t\right)\right]+b_{i}\left(P_{i}\left(t\right)-P_{d}\left(t\right)-\Delta_{i0}\left(t\right)\right)\;\;$$
(9)$$e_{iV}=\sum_{j\in N_{i}}a_{ij}\left[V_{i}\left(t\right)-V_{j}\left(t\right)-{\dot{\Delta }}_{ij}\left(t\right)\right]+b_{i}\left(V_{i}\left(t\right)-{\dot{P}}_{d}\left(t\right)-{\dot{\Delta }}_{i0}\left(t\right)\right)\;\;$$
where $\Delta_{ij}=\Lambda_{i}-\Lambda_{j}$, $\Delta_{i0}=\Lambda_{i}-\Lambda_{0}$.

Then the error dynamics of system (2) can be expressed in a compact form as follows
(10)$$\left\{ \begin{array}{c} {\dot{e}}_{P}=e_{V}\\ {\dot{e}}_{V}=\left(\left(L+B\right)⊗I_{3\times 3}\right)\left(U_{P}+F_{P}-e_{n}⊗{\ddot{P}}_{d}-{\ddot{\Delta }}_{\Sigma }\right) \end{array}\;\;\right.$$
where $e_{P}={\left[e_{1P}^{T},e_{2P}^{T},...,e_{NP}^{T}\right]}^{T}$, $e_{V}={\left[e_{1V}^{T},e_{2V}^{T},...,e_{NV}^{T}\right]}^{T}$, $U_{P}={\left[U_{1P}^{T},U_{2P}^{T},...,U_{NP}^{T}\right]}^{T}$, $U_{iP}=-ge_{3}+R_{ti}\frac{T_{i}}{m_{i}}e_{3}$, $F_{P}={\left[F_{1P}^{T},F_{2P}^{T},...,F_{NP}^{T}\right]}^{T}$, $\Delta_{\Sigma }={\left[\Delta_{10}^{T},\Delta_{20}^{T},...,\Delta_{N0}^{T}\right]}^{T}$, $e_{n}={\left[1,1,...,1\right]}^{T}\in R^{n}$.

**Assumption** **3.**
*The second derivatives of $P_{d}$ and $\Delta_{\Sigma }$ are bounded; there exists positive constants $P_{M}$ and $\Delta_{M}$, such that $∥e_{n}⊗{\ddot{P}}_{d}∥\leq P_{M}$ and $∥{\ddot{\Delta }}_{\Sigma }∥\leq \Delta_{M}$.*


To obtain the approximation of the lumped uncertainty $F_{iP}$, we adopt an adaptive RBFNN with time-delayed states $P_{i}\left(t-\tau_{i}\right)$ and $V_{i}\left(t-\tau_{i}\right)$ as inputs and approximation value as output, which is expressed as
(11)$${\hat{F}}_{iP}={\hat{W}}_{i}^{T}\Psi_{i}\;\;$$
where ${\hat{W}}_{i}=diag\{{\hat{W}}_{i1},{\hat{W}}_{i2},{\hat{W}}_{i3}\}$ is the current RBFNN weights estimation value of *i*-th QR, $\Psi_{i}={\left[\Psi_{i1}^{T},\Psi_{i2}^{T},\Psi_{i3}^{T}\right]}^{T}$, ${\hat{W}}_{ik}\in R^{l_{ik}}$, $\Psi_{ik}\in R^{l_{ik}}$, k=1,2,3.

Then $F_{P}$ can be expressed as
(12)$$F_{P}=W^{*T}\Psi +\epsilon \;\;$$
where $W^{*}=diag\{W_{1}^{*},W_{2}^{*},...,W_{N}^{*}\}$, $\Psi ={\left[\Psi_{1}^{T},\Psi_{2}^{T},...,\Psi_{N}^{T}\right]}^{T}$, $\epsilon ={\left[\epsilon_{1}^{T},\epsilon_{2}^{T},...,\epsilon_{N}^{T}\right]}^{T}$ with $\epsilon_{i}={\left[\epsilon_{i1},\epsilon_{i2},\epsilon_{i3}\right]}^{T}$, the approximation of $F_{P}$ is
(13)$${\hat{F}}_{P}={\hat{W}}^{T}\Psi \;\;$$
where $\hat{W}=diag\{{\hat{W}}_{1},{\hat{W}}_{2},...,{\hat{W}}_{N}\}$.

In addition, the RBFNN weights estimation error is denoted as $\tilde{W}=W^{*}-\hat{W}$.

**Remark** **1.**
*In the light of Stone–Weierstrass approximation theorem [44], Ψ, $W^{*}$ and ϵ are bounded, namely, $∥\Psi ∥\leq \Psi_{M}$, $∥W^{*}∥\leq W_{M}$ and $∥\epsilon ∥\leq \epsilon_{M}$, $\Psi_{M}$, $W_{M}$ and $\epsilon_{M}$ are positive numbers.*


Now we design the control inputs $U_{iP}$ of *i*-th QR position subsystem (2) and update laws of RBFNN weights ${\hat{W}}_{i}$ as:(14)$$U_{iP}=-k_{P}e_{iP\tau }-k_{V}e_{iV\tau }-{\hat{W}}_{i}^{T}\Psi_{i}+{\ddot{\Delta }}_{i0\tau }\;\;$$
(15)$${\dot{\hat{W}}}_{i}=\Pi_{i}\left(\Psi_{i}^{*}\left(K_{P}e_{iP\tau }+K_{V}e_{iV\tau }\right)-K_{W}{\hat{W}}_{i}\right)\;\;$$
where $k_{P},k_{V}>0$. $\Delta_{i0\tau }=\Delta_{i0}\left(t-\tau_{i}\right)$, $e_{iP\tau }=e_{iP}\left(t-\tau_{i}\right)$ and $e_{iV\tau }=e_{iV}\left(t-\tau_{i}\right)$. $K_{P},K_{V}$ and $K_{W}$ are positive design constants, $\Psi_{i}^{*}=diag\{\Psi_{i1},\Psi_{i2},\Psi_{i3}\}$, $\Pi_{i}=diag\{\Pi_{i1},\Pi_{i2},\Pi_{i3}\}$, $\Pi_{ik}=\kappa_{ik}I_{l_{ik}\times l_{ik}}$ is positive definite with $\kappa_{ik}>0$, k=1,2,3.

Combining (14) and (15), we have
(16)$${\dot{e}}_{V}=\left(\left(L+B\right)⊗I_{3\times 3}\right)\left(-k_{P}e_{P\tau }-k_{V}e_{V\tau }+{\tilde{W}}^{T}\Psi +\epsilon -e_{n}⊗{\ddot{P}}_{d}+{\ddot{\Delta }}_{\Sigma \tau }-{\ddot{\Delta }}_{\Sigma }\right)\;\;$$
where $e_{P\tau }=e_{P}\left(t-\tau_{i}\right)$, $e_{V\tau }=e_{V}\left(t-\tau_{i}\right)$ and $\Delta_{\Sigma \tau }=\Delta_{\Sigma }\left(t-\tau_{i}\right)$.

**Lemma** **1.**
*Under Assumption 1, $G=\left(L+B\right)⊗I_{3\times 3}$ is positive definite [45], so $∥\Delta_{P}∥\leq ∥e_{P}∥\sigma_{min}^{-1}\left(G\right)$ and $∥\Delta_{V}∥\leq ∥e_{V}∥\sigma_{min}^{-1}\left(G\right)$ [44], in which $\Delta_{P}$ and $\Delta_{V}$ are formation tracking errors, $\Delta_{P}=P-e_{n}⊗P_{d}-\Delta_{\Sigma }$ with $P={\left[P_{1},P_{2},...,P_{N}\right]}^{T}$, $\Delta_{V}={\dot{\Delta }}_{P}$.*


**Lemma** **2.**
*According to [46], we can conclude that the following inequality is always valid:*

(17)
$$\tau_{0}^{-1}{\left[h\left(t\right)-h\left(t-\tau \right)\right]}^{T}U\left[h\left(t\right)-h\left(t-\tau \right)\right]\leq \int_{t-\tau }^{t}{\dot{h}}^{T}\left(\xi \right)U\dot{h}\left(\xi \right)d\xi \leq \int_{t-\tau_{0}}^{t}{\dot{h}}^{T}\left(\xi \right)U\dot{h}\left(\xi \right)d\xi \;\;$$

*where t>0, $h\left(t\right)\in R^{n}$ and $\tau \left(t\right)\in \left[0,\tau_{0}\right]$ are arbitrary differentiable vector and scalar functions, respectively, and $\tau_{0}>0$. $U=U^{T}$ is an arbitrary positive definite constant matrix.*


**Theorem** **1.**
*Under Assumptions 1–3, with the control law (14) and update law (15), the time-varying formation tracking for N QRs position systems (2) subject to time-varying delays and uncertainties can be achieved if the positive design constants $K_{P}=4M_{2}\tau_{M}^{2}k_{v},K_{V}=4M_{2}\tau_{M}^{2}k_{p},K_{0}=4M_{2}\tau_{M}^{2}k_{P}k_{V}$ and $k_{P}$, $k_{V}$, $K_{W}$, $M_{1}$, $M_{2}$ are chosen appropriately to make the symmetric matrix M be positive definite, which is*

(18)
$$\begin{array}{c} M=[\begin{array}{ccccc} k_{P}K_{V}-2M_{2}\tau_{M}^{2}k_{P}^{2} & 0 & 0 & 0 & 0\\ 0 & k_{V}K_{P}-M_{1}\tau_{M}^{2}-K_{V}\sigma_{max}\left(G^{-1}\right)-2M_{2}\tau_{M}^{2}k_{V}^{2} & 0 & 0 & 0\\ 0 & 0 & M_{1}-2M_{2}\tau_{M}^{2}k_{P}^{2} & -2M_{2}\tau_{M}^{2}k_{P}k_{V} & -\frac{K_{V}}{2}\Psi_{M}\\ 0 & 0 & 0 & M_{2}\left(\sigma_{min}\left(B^{-1}\right)-2\tau_{M}^{2}k_{V}^{2}\right) & -\frac{K_{P}}{2}\Psi_{M}\\ 0 & 0 & 0 & 0 & K_{W}-2M_{2}\tau_{M}^{2}\Psi_{M}^{2} \end{array}] \end{array}\;\;$$

*where $B=G^{T}G$.*


**Proof.** Consider the Lyapunov–Krasovskii candidate function as $V\left(t\right)=V_{1}\left(t\right)+V_{2}\left(t\right)+V_{3}\left(t\right)+V_{4}\left(t\right)$ with
(19)$$V_{1}\left(t\right)=\frac{1}{2}K_{P}e_{V}^{T}G^{-1}e_{V}+\frac{1}{2}K_{0}e_{P}^{T}e_{P}+K_{V}e_{P}^{T}G^{-1}e_{V}\;\;$$
(20)$$V_{2}\left(t\right)=M_{1}\tau_{M}\int_{t-\tau_{M}}^{t}\left(\xi -t+\tau_{M}\right){\dot{e}}_{P}^{T}\left(\xi \right){\dot{e}}_{P}\left(\xi \right)d\xi \;\;$$
(21)$$V_{3}\left(t\right)=M_{2}\tau_{M}\int_{t-\tau_{M}}^{t}\left(\xi -t+\tau_{M}\right){\dot{e}}_{V}^{T}\left(\xi \right)B^{-1}{\dot{e}}_{V}\left(\xi \right)d\xi \;\;$$
(22)$$V_{4}=tr\{{\tilde{W}}^{T}\Pi^{-1}\tilde{W}\}\;\;$$
where $\Pi =diag\{\Pi_{1},\Pi_{2},...,\Pi_{N}\}$.Taking the time derivative of $V_{1}$ and $V_{4}$ we have
(23)$$\begin{array}{cc} {\dot{V}}_{1}+{\dot{V}}_{4}= & \left(K_{P}e_{V}^{T}+K_{V}e_{P}^{T}\right)G^{-1}{\dot{e}}_{V}+K_{0}e_{P}^{T}e_{V}+e_{V}^{T}K_{V}G^{-1}e_{V}+tr\{{\tilde{W}}^{T}\Pi^{-1}\dot{\tilde{W}}\}\\ = & \left(K_{P}e_{V}^{T}+K_{V}e_{P}^{T}\right)\left(-k_{P}\Delta e_{P}-k_{P}e_{P}-k_{V}\Delta e_{V}-k_{V}e_{V}\right)+K_{0}e_{P}^{T}e_{V}\\ & +\left(K_{P}e_{V}^{T}+K_{V}e_{P}^{T}\right)H+e_{V}^{T}K_{V}G^{-1}e_{V}+tr\{{\tilde{W}}^{T}K_{W}\hat{W}\}\\ & +tr\{{\tilde{W}}^{T}\Psi [\left(K_{P}e_{V}^{T}+K_{V}e_{P}^{T}\right)-\left(K_{P}e_{V\tau }^{T}+K_{V}e_{P\tau }^{T}\right)]\}\\ = & \left(K_{P}e_{V}^{T}+K_{V}e_{P}^{T}\right)\left(-k_{P}\Delta e_{P}-k_{P}e_{P}-k_{V}\Delta e_{V}-k_{V}e_{V}\right)+K_{W}W_{M}tr{∥\tilde{W}∥}_{F}\\ & +\left(K_{P}e_{V}^{T}+K_{V}e_{P}^{T}\right)H+K_{0}e_{P}^{T}e_{V}^{T}+e_{V}^{T}K_{V}G^{-1}e_{V}-K_{W}{∥\tilde{W}∥}_{F}^{2}\\ & +tr\{{\tilde{W}}^{T}\Psi [\left(K_{P}e_{V}^{T}+K_{V}e_{P}^{T}\right)-\left(K_{V}e_{P}^{T}+K_{V}\Delta e_{P}^{T}+K_{P}e_{V}^{T}+K_{P}\Delta e_{V}^{T}\right)]\} \end{array}\;\;$$
where $\Delta e_{P}=e_{P\tau }-e_{P}$, $\Delta e_{V}=e_{V\tau }-e_{V}$ and $H=\epsilon -e_{n}⊗{\ddot{P}}_{d}+{\ddot{\Delta }}_{\Sigma \tau }-{\ddot{\Delta }}_{\Sigma }$.By Lemma 2, we obtain the time derivatives of $V_{2}$ and $V_{3}$ as follows
(24)$$\begin{array}{cc} {\dot{V}}_{2} & =M_{1}\tau_{M}^{2}{\dot{e}}_{P}^{T}\left(t\right){\dot{e}}_{P}\left(t\right)-M_{1}\tau_{M}\int_{t-\tau_{M}}^{t}{\dot{e}}_{P}^{T}\left(\xi \right){\dot{e}}_{P}\left(\xi \right)d\xi \\ & \leq -M_{1}\Delta e_{P}^{T}\Delta e_{P}+M_{1}\tau_{M}^{2}e_{V}^{T}\left(t\right)e_{V}\left(t\right)\\ & \leq -M_{1}{∥\Delta e_{P}∥}^{2}+M_{1}\tau_{M}^{2}{∥e_{V}∥}^{2} \end{array}\;\;$$
and
(25)$$\begin{array}{cc} {\dot{V}}_{3}= & M_{2}\tau_{M}^{2}{\dot{e}}_{V}^{T}\left(t\right)B^{-1}{\dot{e}}_{V}\left(t\right)-M_{2}\tau_{M}\int_{t-\tau_{M}}^{t}{\dot{e}}_{V}^{T}\left(\xi \right)B^{-1}{\dot{e}}_{V}\left(\xi \right)d\xi \\ \leq & -M_{2}\Delta e_{V}^{T}B^{-1}\Delta e_{V}+M_{2}\tau_{M}^{2}{\dot{e}}_{V}^{T}\left(t\right)B^{-1}{\dot{e}}_{V}\left(t\right)\\ \leq & -M_{2}\sigma_{min}\left(B^{-1}\right){∥\Delta e_{V}∥}^{2}+M_{2}\tau_{M}^{2}(-k_{P}\Delta e_{P}-k_{P}e_{P}-k_{V}\Delta e_{V}-k_{V}e_{V}\\ & +{\tilde{W}}^{T}{\Psi +H)}^{T}\left(-k_{P}\Delta e_{P}-k_{P}e_{P}-k_{V}\Delta e_{V}-k_{V}e_{V}+{\tilde{W}}^{T}\Psi +H\right)\\ \leq & -M_{2}\sigma_{min}\left(B^{-1}\right){∥\Delta e_{V}∥}^{2}+2M_{2}\tau_{M}^{2}{∥\tilde{W}∥}_{F}^{2}\Psi_{M}^{2}+2M_{2}\tau_{M}^{2}(-k_{P}\Delta e_{P}-k_{P}e_{P}\\ & -k_{V}\Delta e_{V}-k_{V}e_{V}{+H)}^{T}\left(-k_{P}\Delta e_{P}-k_{P}e_{P}-k_{V}\Delta e_{V}-k_{V}e_{V}+H\right)\\ \leq & -M_{2}\sigma_{min}\left(B^{-1}\right){∥\Delta e_{V}∥}^{2}+M_{2}\tau_{M}^{2}(2k_{P}^{2}{∥\Delta e_{P}∥}^{2}+4k_{P}^{2}e_{P}^{T}\Delta e_{P}\\ & +4k_{P}k_{V}\Delta e_{V}^{T}\Delta e_{P}+4k_{P}k_{V}e_{V}^{T}\Delta e_{P}+4k_{P}k_{V}e_{V}^{T}e_{P}+2k_{P}^{2}{∥e_{P}∥}^{2}\\ & +4k_{P}k_{V}\Delta e_{V}^{T}e_{P}+2k_{V}^{2}{∥\Delta e_{V}∥}^{2}+4k_{V}^{2}e_{V}^{T}\Delta e_{V}+2k_{P}^{2}{∥e_{V}∥}^{2}\\ & +2H_{M}^{2}+2∥\tilde{W}{∥}_{F}^{2}\Psi_{M}+4H^{T}\left(-k_{P}\Delta e_{P}-k_{P}e_{P}-k_{V}\Delta e_{V}-k_{V}e_{V}\right)) \end{array}\;\;$$
where $H_{M}=2\Delta_{M}+P_{M}+\epsilon_{M}$. By applying (23)–(25) we obtain
(26)$$\dot{V}={\dot{V}}_{1}+{\dot{V}}_{2}+{\dot{V}}_{3}+{\dot{V}}_{4}\leq -e^{T}Me+e^{T}z+\Theta =-V_{e}\left(e\right)\;\;$$
where $e={\left[∥e_{P}∥,∥e_{V}∥,∥\Delta e_{P}∥,∥\Delta e_{V}∥,{∥\tilde{W}∥}_{F}\right]}^{T}$, $z={\left[0,0,4k_{P}H_{M},4k_{V}H_{M},W_{M}K_{W}\right]}^{T}$, and $\Theta =2M_{2}\tau_{M}^{2}H_{M}^{2}$, M are defined by (18). If M is positive definite, then $V_{e}\left(e\right)>0$, and
(27)$$∥e∥\geq \frac{-∥z∥+\sqrt{{∥z∥}^{2}+4\sigma_{min}\left(M\right)\Theta }}{2\sigma_{min}\left(M\right)}\;\;$$Thus, e is uniformly ultimately bounded (UUB) according to [47]. Moreover, $e_{P}$, $e_{V}$ are bounded stable referring to the definition of e, and following Lemma 1, the formation tracking errors $\Delta_{P}$ and $\Delta_{V}$ are also UUB. So, the desired position control for formation flight can be realized by control law (14) and RBFNN update law (15). □

**Remark** **2.**
*In matrix M, $k_{p}$, $k_{v}$, $K_{P}$, $K_{V}$ and $K_{W}$ are control and adaptive parameters, and $K_{0}$, $M_{1}$ and $M_{2}$ are constants to be selected. $\tau_{M}$ is the upper bound of time delays. Except $\tau_{M}$, all of the above parameters are adjustable to ensure the solvability of M. Besides, one can see that all of the diagonal elements of M are positive when $\tau_{M}$ being a certain value. Therefore, M is solvable in Theorem 1.*


### 3.3. Design of AFTAC

The command attitude $A_{iC}\left(t\right)={\left[\phi_{iC},\theta_{iC},\psi_{iC}\right]}^{T}$ and total thrust $T_{iC}$ of *i*-th QR can be obtained from $U_{iP}={\left[U_{i1P},U_{i2P},U_{i3P}\right]}^{T}$, which is derived as
(28)$$\left\{ \begin{array}{c} T_{iC}=m_{i}\sqrt{U_{i1P}^{2}+U_{i2P}^{2}+{\left(U_{i3P}+g\right)}^{2}}\\ \phi_{iC}=arcsin\left(\frac{U_{i1P}sin\psi_{iC}-U_{i2P}cos\psi_{iC}}{m_{i}^{-1}T_{i}}\right)\\ \theta_{iC}=arctan\left(\frac{U_{i1P}cos\psi_{iC}+U_{i2P}sin\psi_{iC}}{U_{i3P}+g}\right) \end{array}\;\;\right.$$
where $\psi_{iC}$ is a free variable and can be set to $\psi_{iC}=0$ for simplicity.

**Remark** **3.**
*It is feasible to ensure $U_{i3P}+g$ is constantly positive to avoid singularity because $U_{i3P}$ is bounded by selecting suitable gain constant $k_{P},k_{V}$ and ${\hat{W}}_{i}^{T}\Psi_{i},{\ddot{\Delta }}_{i0\tau }$ is in a cetain range when calculating $\theta_{iC}$ in (28).*


To deploy the attitude control scheme, the following assumptions and lemma need to be made:

**Assumption** **4.**
*There exists a 2nd order differentiable continuous bound $C_{id}\left(t\right)\in R$ of command attitude $A_{iC}$ within the constraint $C_{i1}$, namely, $∥A_{iC}\left(t\right)∥\leq C_{id}\left(t\right)<{\bar{C}}_{i1}\left(t\right)$. The initial state of the attitude subsystem needs to be within the constraints ${\bar{C}}_{im},m=1,2$, which is the (3−m)-th order differentiable.*


**Assumption** **5.**
*The actuators will not completely fail during operation, and the fault signals $\Gamma_{ik}\left(t\right)$ and $\delta_{i}\left(t\right)$ change continuously within certain ranges, that is, $0<\Gamma_{ik,min}\leq \Gamma_{ik}\left(t\right)\leq \Gamma_{ik,max}$, $tr({\dot{\Gamma }}_{i}^{T}{\dot{\Gamma }}_{i})\leq \Xi_{i}<\infty$, where $\Gamma_{ik,min}$, $\Gamma_{ik,max}$ are known constants with k=1,2,3, and $∥\delta_{i}\left(t\right)∥\leq {\bar{\delta }}_{i}<\infty ,\;∥{\dot{\delta }}_{i}\left(t\right)∥\leq {\bar{\delta }}_{i0}<\infty$.*


**Assumption** **6.**
*The model uncertainty factor $\Delta F_{i}\left(t\right)$ and its derivatives and unknown disturbances $D_{iA}$ are bounded, which are expressed as $tr(\Delta F_{i}^{T}\Delta F_{i})\leq \Xi_{i}<\infty$, $tr(\Delta {\dot{F}}_{i}^{T}\Delta {\dot{F}}_{i})\leq {\bar{\Xi }}_{i}$, $∥D_{iA}∥\leq {\bar{D}}_{iA}$, ${\bar{D}}_{iA}>0\in R$ can be unknown.*


**Lemma** **3.**
*By [48], we know that the following inequality holds:*

(29)
$$0\leq |\rho |-\rho tanh\left(\frac{\rho }{\lambda }\right)\leq \lambda \kappa_{0}\;\;$$

*where λ>0,ρ∈R are arbitrary numbers, $\kappa_{0}=0.2785$.*


The attitude tracking error of *i*-th QR is $z_{i1}=A_{i}-A_{iC}$, of which the dynamic can be derived as
(30)$${\dot{z}}_{i1}=R_{ir}\left(z_{i2}+\alpha_{i}\right)-{\dot{A}}_{iC}\;\;$$
where $z_{i2}=\Omega_{i}-\alpha_{i}$ is angular velocity tracking error, $\alpha_{i}$ is the command filtered signal of designed virtual control law $\alpha_{iC}$, in which the command filter limits the magnitude, rate and bandwidth of $\alpha_{iC}$ and is shown in Figure 3.

In order to deal with the constraints on the attitude states, we adopt the tan-type BLF as follows
(31)$$V_{i1B}=\frac{C_{i1}^{2}}{\pi }\tan(\frac{\pi z_{i1}^{T}z_{i1}}{2C_{i1}^{2}})\;\;$$
where $C_{i1}={\bar{C}}_{i1}-C_{id}$. It is easy to see that when $∥z_{im}∥\to C_{im}$, then $V_{imB}\to \infty$; thus, $∥z_{im}∥<C_{im}$ holds if and only if $V_{imB}$ is bounded, m=1,2.

**Remark** **4.**
*When there is no attitude constraint on i-th QR, then ${\bar{C}}_{im}\to \infty$; thus, $C_{im}\to \infty ,m=1,2$, and we have*

(32)
$$\lim_{C_{im}\to \infty }V_{imB}=\frac{1}{2}z_{im}^{T}z_{im}\;\;$$

*that is, our BLF analysis method is also available for the unconstrained circumstance.*


For simplicity of notation, define $\nu_{im}=\frac{z_{im}}{\cos^{2}(\frac{\pi z_{im}^{T}z_{im}}{2C_{im}^{2}})},m=1,2$ and take the derivative of $V_{i1B}$ with respect to time, and we have
(33)$${\dot{V}}_{i1B}=\frac{2C_{i1}{\dot{C}}_{i1}}{\pi }\tan(\frac{\pi z_{i1}^{T}z_{i1}}{2C_{i1}^{2}})+\nu_{i1}^{T}(R_{ir}(z_{i2}+\alpha_{i})-{\dot{A}}_{iC})-(\frac{{\dot{C}}_{i1}}{C_{i1}})\nu_{i1}^{T}z_{i1}\;\;$$

The designed virtual control law $\alpha_{iC}$ is shown as below
(34)$$\alpha_{iC}=R_{ir}^{-1}(-K_{i1\alpha }\frac{z_{i1}}{z_{i1}^{T}z_{i1}}\frac{C_{i1}^{2}}{\pi }\sin(\frac{\pi z_{i1}^{T}z_{i1}}{2C_{i1}^{2}})\cos(\frac{\pi z_{i1}^{T}z_{i1}}{2C_{i1}^{2}})+K_{i1\alpha }E_{i1}-K_{i1C}z_{i1}+{\dot{A}}_{iC}-\frac{K_{i1\alpha }^{2}}{2}\nu_{i1})\;\;$$
where $\mu_{i1}$ is a positive small constant, $K_{i1\alpha }>2K_{i1C}>0$, $K_{i1\alpha }$ is a design parameter, and $K_{i1C}=\sqrt{{\left(\frac{{\dot{C}}_{i1}}{C_{i1}}\right)}^{2}+\epsilon_{i1C}}$ with the small constant $\epsilon_{i1C}>0$.

**Remark** **5.**
*To make ${\dot{V}}_{i1B}$ be negative definite, the terms $-(\frac{{\dot{C}}_{i1}}{C_{i1}})\nu_{i1}^{T}z_{i1}$, $\nu_{i1}^{T}{\dot{A}}_{iC}$, $\frac{K_{i1\alpha }^{2}}{2}\nu_{i1}^{T}\nu_{i1}$ in (33) will be canceled by terms $-K_{i1C}z_{i1}$, ${\dot{A}}_{iC}$, $-\frac{K_{i1\alpha }^{2}}{2}\nu_{i1}$ in (34), respectively. Noticing that*

$$-K_{i1\alpha }\nu_{i1}^{T}\frac{z_{i1}}{z_{i1}^{T}z_{i1}}\frac{C_{i1}^{2}}{\pi }\sin(\frac{\pi z_{i1}^{T}z_{i1}}{2C_{i1}^{2}})\cos(\frac{\pi z_{i1}^{T}z_{i1}}{2C_{i1}^{2}})=-K_{i1\alpha }\frac{k_{i1\alpha }^{2}}{\pi }\tan(\frac{\pi z_{i1}^{T}z_{i1}}{2C_{i1}^{2}}),$$

*this will generate the negative definite BLF-form term in (33).*


The auxiliary system $E_{i1}$ is designed as
(35)$${\dot{E}}_{i1}=\{\begin{array}{cc} -K_{i1E}E_{i1}-\varkappa_{i1}E_{i1}+\gamma_{i1E}\Delta \alpha_{i}, & if\;∥E_{i1}∥>{\bar{E}}_{i1}\\ 0, & if\;∥E_{i1}∥\leq {\bar{E}}_{i1} \end{array}\;\;$$
where $E_{i1}\in R^{3}$, ${\bar{E}}_{i1}>0$ is a small constant, $K_{i1E}>1$, $\gamma_{i1E}>0$ are design constants, $\varkappa_{i1}=\frac{|\nu_{1}^{T}R_{ir}\Delta \alpha_{i1}|+\frac{1}{2}\gamma_{i1E}^{2}\Delta \alpha_{i}^{T}\Delta \alpha_{i}}{{∥E_{i1}∥}^{2}}$, $\Delta \alpha_{i}=\alpha_{i}-\alpha_{iC}$.

**Remark** **6.**
*When saturation occurs, the auxiliary system will respond to it. Otherwise, $\Delta \alpha_{i}=0$, then ${\dot{E}}_{i1}=-K_{i1E}E_{i1}$; thus, $E_{i1}$ will converge into $∥E_{i1}∥\leq {\bar{E}}_{i1}$, after which, if saturation occurs again, $E_{i1}$ can be reset so that $∥E_{i1}∥>{\bar{E}}_{i1}$. Then, the auxiliary system can be made responsive again.*


As can be seen from (33),
$$\frac{2C_{i1}{\dot{C}}_{i1}}{\pi }\tan(\frac{\pi z_{i1}^{T}z_{i1}}{2C_{i1}^{2}})=\frac{{2\dot{C}}_{i1}}{C_{i1}}\frac{C_{i1}^{2}}{\pi }\tan(\frac{\pi z_{i1}^{T}z_{i1}}{2C_{i1}^{2}})<\frac{2K_{i1C}C_{i1}^{2}}{\pi }\tan(\frac{\pi z_{i1}^{T}z_{i1}}{2C_{i1}^{2}}).$$

Besides, $\nu_{i1}^{T}K_{i1\alpha }E_{i1}\leq \frac{K_{i1\alpha }^{2}}{2}\nu_{i1}^{T}\nu_{i1}+\frac{1}{2}E_{i1}^{T}E_{i1}$. Set $K_{i1}^{*}=K_{i1\alpha }-2K_{i1C}$, then we have
(36)$${\dot{V}}_{i1B}\leq -K_{i1}^{*}\frac{C_{i1}^{2}}{\pi }\tan(\frac{\pi z_{i1}^{T}z_{i1}}{2C_{i1}^{2}})+\nu_{i1}^{T}R_{ir}z_{i2}+\frac{1}{2}E_{i1}^{T}E_{i1}+|\nu_{i1}^{T}R_{ir}\Delta \alpha_{i}|\;\;$$

The Lyapunov function for this step is constructed as
(37)$$V_{i1}^{*}=V_{i1B}+\frac{1}{2}E_{i1}^{T}E_{i1}\;\;$$

Taking the time derivative of (37) and notice that
(38)$$E_{i1}^{T}\gamma_{i1E}\Delta \alpha_{i}\leq \frac{\gamma_{i1E}^{2}}{2}\Delta \alpha_{i}^{T}\Delta \alpha_{i}+\frac{1}{2}E_{i1}^{T}E_{i1}\;\;$$
then we have
(39)$${\dot{V}}_{i1}^{*}\leq -K_{i1}^{*}\frac{C_{i1}^{2}}{\pi }\tan(\frac{\pi z_{i1}^{T}z_{i1}}{2C_{i1}^{2}})-(K_{i1E}-1)E_{i1}^{T}E_{i1}+\nu_{i1}^{T}R_{ir}z_{i2}\;\;$$
where $\nu_{i1}^{T}R_{ir}z_{i2}$ will be compensated later.

Similarly, the BLF for m=2 is
(40)$$V_{i2B}=\frac{C_{i2}^{2}}{\pi }\tan(\frac{\pi z_{i2}^{T}z_{i2}}{2C_{i2}^{2}})\;\;$$
where $C_{i2}={\bar{C}}_{i2}-{\bar{\alpha }}_{i}>0$, and $∥\alpha_{i}∥<{\bar{\alpha }}_{i}$

The dynamics of the angular velocity tracking error $z_{i2}$ is derived as
(41)$${\dot{z}}_{i2}={\Delta F}_{i}F_{iA}+J_{i}^{-1}(\Delta U_{iA}+U_{iAF})+D_{iA}-{\dot{\alpha }}_{i}\;\;$$
where $\Delta U_{iA}=U_{iA}-U_{iAF}$. According to (4), the following inequality holds:(42)$$∥U_{iA}-U_{iAF}∥\leq ∥U_{iA}∥+∥\Gamma_{i}∥∥U_{iAC}∥+∥\delta_{i}∥\leq ∥{\bar{U}}_{iA}∥\;\;$$
where $∥U_{iA}∥$ is the given input, $∥U_{iAC}∥$ is produced by our desired control law and $∥\Gamma_{i}∥$ and $∥\delta_{i}∥$ are known from Assumption 5; thus, ${\bar{U}}_{iA}$ can be determined. Define $U_{iAC}$ as
(43)$$U_{iAC}={{\hat{\Gamma }}_{i}}^{-1}\Phi_{i}\;\;$$
where ${\hat{\Gamma }}_{i}$ is the estimation of the multiplicative fault $\Gamma_{i}$, $\Phi_{i}$ will be designed later. Notice that
(44)$$\begin{array}{cc} J_{i}^{-1}U_{iAF} & =J_{i}^{-1}(\Gamma_{i}U_{iAC}+\delta_{i})\\ & =J_{i}^{-1}\delta_{i}+J_{i}^{-1}({\hat{\Gamma }}_{i}{{\hat{\Gamma }}_{i}}^{-1}\Phi_{i}-{\hat{\Gamma }}_{i}{{\hat{\Gamma }}_{i}}^{-1}\Phi_{i}+\Gamma_{i}U_{iAC})\\ & =J_{i}^{-1}\delta_{i}-J_{i}^{-1}{\tilde{\Gamma }}_{i}U_{iAC}+J_{i}^{-1}\Phi_{i} \end{array}\;\;$$
where ${\tilde{\Gamma }}_{i}={\hat{\Gamma }}_{i}-\Gamma_{i}$ is the estimation error of multiplicative fault $\Gamma_{i}$. Then, the desired control law $\Phi_{i}$ is designed as
(45)$$\begin{array}{cc} \Phi_{i}= & -{\hat{\delta }}_{i}+J_{i}(-K_{i2C}z_{i2}-\cos^{2}(\frac{\pi z_{i2}^{T}z_{i2}}{2C_{i2}^{2}})R_{ir}^{T}\nu_{i1}+{\dot{\alpha }}_{i}-\Delta {\hat{F}}_{i}F_{iA}-\frac{K_{i2\Phi }^{2}}{2}\nu_{i2}+K_{i2\Phi }E_{i2}\\ & -{\hat{D}}_{iA}\tanh(\frac{\nu_{i2}}{\mu_{i2}})-K_{i2\Phi }\frac{z_{i2}}{z_{i2}^{T}z_{i2}}\frac{C_{i2}^{2}}{\pi }\sin(\frac{\pi z_{i2}^{T}z_{i2}}{2C_{i2}^{2}})\cos(\frac{\pi z_{i2}^{T}z_{i2}}{2C_{i2}^{2}})) \end{array}\;\;$$
where $\epsilon_{i2C}>0$, $\mu_{i2}>0$ are small constants, $K_{i2\Phi }>0$ is a design parameter and $K_{i2\Phi }>2K_{i2C}$ with $K_{i2C}=\sqrt{{\left(\frac{{\dot{C}}_{i2}}{C_{i2}}\right)}^{2}+\epsilon_{i2C}}$, ${\hat{D}}_{iA}$ is the estimation value of disturbances’ upper bound ${\bar{D}}_{iA}$.

The update law for $\Delta {\hat{F}}_{i}$, ${\hat{\delta }}_{i}$, ${\hat{\Gamma }}_{i}$ and ${\hat{D}}_{iA}$ are constructed as
(46)$$\Delta {\dot{\hat{F}}}_{i}=\gamma_{iA}F_{iA}\nu_{i2}^{T}-\beta_{iA}\Delta {\hat{F}}_{i}\;\;$$
(47)$${\dot{\hat{\delta }}}_{i}=\kappa_{i1}J_{i}^{-T}\nu_{i2}-\eta_{i}{\hat{\delta }}_{i}\;\;$$
(48)$${\dot{\hat{\Gamma }}}_{ik}=\{\begin{array}{cc} 0, & if\;{\hat{\Gamma }}_{ik}={\hat{\Gamma }}_{ik,min},\;and,\;\chi_{ik}\left(t\right)<0\\ \chi_{ik}\left(t\right), & else \end{array}\;\;$$
(49)$${\dot{\hat{D}}}_{iA}=\xi_{iA}(\nu_{i2}^{T}\tanh(\frac{\nu_{i2}}{\mu_{i2}})-\lambda_{iA}{\hat{D}}_{iA})\;\;$$
where $\chi_{ik}\left(t\right)=\kappa_{i2}\nu_{i2}^{T}J_{i}^{-1}U_{iAC}-\Upsilon_{i}{\hat{\Gamma }}_{ik}$, k=1,2,3, $\gamma_{i2}>0$, $\beta_{i2}>1$, $\eta_{i}>1$, $\kappa_{i1}>0$, $\kappa_{i2}>0$, Υ>1, $\xi_{iA}>0$, $\lambda_{iA}>0$ are design constants.

Similarly, auxiliary system $E_{i2}$ is designed as
(50)$${\dot{E}}_{i2}=\{\begin{array}{cc} -K_{i2E}E_{i2}-\varkappa_{i2}E_{i2}+\gamma_{i2E}{\bar{U}}_{iA}, & if\;∥E_{i2}∥>{\bar{E}}_{i2}\\ 0, & if\;∥E_{i2}∥\leq {\bar{E}}_{i2} \end{array}\;\;$$
where $\varkappa_{i2}=\frac{∥\nu_{i2}^{T}J_{i}^{-1}∥∥{\bar{U}}_{iA}∥+\frac{1}{2}\gamma_{i2E}^{2}{∥{\bar{U}}_{iA}∥}^{2}}{{∥E_{i2}∥}^{2}}$, $K_{i2E}>1$, $\gamma_{i2E}>0$ are design parameters and ${\bar{E}}_{i2}>0$ is a small constant.

Set $K_{i2}^{*}=K_{i2\Phi }-2K_{i2C}$, similarly as (36), and we can obtain
(51)$$\begin{array}{cc} {\dot{V}}_{i2B}\leq & -K_{i2}^{*}\frac{C_{i2}^{2}}{\pi }\tan(\frac{\pi z_{i2}^{T}z_{i2}}{2C_{i2}^{2}})+\nu_{i2}^{T}D_{iA}+\frac{1}{2}E_{i2}^{T}E_{i2}-\nu_{i2}^{T}{\hat{D}}_{iA}\tanh(\frac{\nu_{i2}}{\mu_{i2}})\\ & -\nu_{i2}^{T}\Delta {\tilde{F}}_{i}^{T}F_{iA}-\nu_{i2}^{T}J_{i}^{-1}{\tilde{\Gamma }}_{i}U_{iAC}+|\nu_{i2}^{T}J_{i}^{-1}\Delta U_{iA}|-\nu_{i1}^{T}J_{i}^{-1}z_{i2}-\nu_{i2}^{T}J_{i}^{-1}{\tilde{\delta }}_{i} \end{array}\;\;$$
where $\Delta {\tilde{F}}_{i}=\Delta {\hat{F}}_{i}-\Delta F_{i}$ and ${\tilde{\delta }}_{i}={\hat{\delta }}_{i}-\delta_{i}$ are estimation errors of model uncertainty and additive fault, respectively. The Lyapunov candidate function for this step is derived as
(52)$$V_{i2}^{*}=V_{i1}^{*}+V_{i2B}+\frac{1}{2}E_{i2}^{T}E_{i2}+\frac{1}{2\gamma_{i2}}tr(\Delta {\tilde{F}}_{i}^{T}\Delta {\tilde{F}}_{i})+\frac{1}{2\xi_{iA}}{\tilde{D}}_{iA}^{2}+\frac{1}{2\kappa_{i1}}{{\tilde{\delta }}_{i}}^{T}{\tilde{\delta }}_{i}+\frac{1}{2\kappa_{i2}}tr({{\tilde{\Gamma }}_{i}}^{T}{\tilde{\Gamma }}_{i})\;\;$$
where ${\tilde{D}}_{iA}={\hat{D}}_{iA}-D_{iA}$. Taking the time derivative of $V_{i2}^{*}$ and observing that
(53)$$-\lambda_{i1}{\tilde{D}}_{iA}{\hat{D}}_{iA}=-\lambda_{iA}{\tilde{D}}_{iA}({\tilde{D}}_{iA}+{\bar{D}}_{iA})\leq -\frac{\lambda_{iA}}{2}{\tilde{D}}_{iA}^{2}+\frac{\lambda_{iA}}{2}{\bar{D}}_{iA}^{2}\;\;$$
(54)$$\nu_{i2}^{T}D_{iA}-\nu_{i2}^{T}{\hat{D}}_{iA}\tanh(\frac{\nu_{i2}}{\mu_{i2}})+\nu_{i2}^{T}{\tilde{D}}_{iA}\tanh(\frac{\nu_{i2}}{\mu_{i2}})\leq \sum_{k=1}^{3}(|\nu_{ik2}|{\bar{D}}_{iA}-{\bar{D}}_{iA}\nu_{ik2}\tanh(\frac{\nu_{ik2}}{\mu_{i2}}))\leq 3\kappa_{0}{\bar{D}}_{iA}\mu_{i2}\;\;$$
(55)$$\left\{ \begin{array}{c} -\frac{\eta_{i}}{\kappa_{i1}}{{\tilde{\delta }}_{i}}^{T}{\hat{\delta }}_{i}\leq -\frac{\eta_{i}}{2\kappa_{i1}}{{\tilde{\delta }}_{i}}^{T}{\tilde{\delta }}_{i}+\frac{\eta_{i}}{2\kappa_{i1}}{\bar{\delta }}_{i}^{2}\\ -\frac{1}{\kappa_{i1}}{{\tilde{\delta }}_{i}}^{T}{\dot{\delta }}_{i}\leq -\frac{1}{2\kappa_{i1}}{{\tilde{\delta }}_{i}}^{T}{\tilde{\delta }}_{i}+\frac{1}{2\kappa_{i1}}{\bar{\delta }}_{i0}^{2} \end{array}\;\;\right.$$
(56)$$\left\{ \begin{array}{c} -\frac{\Upsilon_{i}}{\kappa_{i2}}tr({{\hat{\Gamma }}_{i}}^{T}{\tilde{\Gamma }}_{i})\leq -\frac{\Upsilon_{i}}{2\kappa_{i2}}tr({{\tilde{\Gamma }}_{i}}^{T}{\tilde{\Gamma }}_{i})+\frac{\Upsilon_{i}}{2\kappa_{i2}}\sum_{k=1}^{3}\Gamma_{ik,max}^{2}\\ \frac{1}{\kappa_{i2}}tr({{\tilde{\Gamma }}_{i}}^{T}{\dot{\Gamma }}_{i})\leq \frac{1}{2\kappa_{i2}}tr({{\tilde{\Gamma }}_{i}}^{T}{\tilde{\Gamma }}_{i})+\frac{1}{2\kappa_{i2}}\Xi_{i} \end{array}\;\;\right.$$
then we have
(57)$$\begin{array}{cc} {\dot{V}}_{i2}^{*}\leq & -\sum_{m=1}^{2}K_{im}^{*}\frac{C_{im}^{2}}{\pi }\tan(\frac{\pi z_{im}^{T}z_{im}}{2C_{im}^{2}})-\frac{\beta_{iA}-1}{2\gamma_{iA}}tr({\Delta \tilde{F}}_{i}^{T}\Delta {\tilde{F}}_{i})-\frac{\lambda_{iA}}{2}{\tilde{D}}_{iA}^{2}-\sum_{m=1}^{2}(K_{imE}-1)E_{im}^{T}E_{im}+S_{i}\\ & -\frac{\Upsilon_{i}-1}{2\kappa_{i2}}tr({{\tilde{\Gamma }}_{i}}^{T}{\tilde{\Gamma }}_{i})-\frac{\eta_{i}-1}{2\kappa_{i1}}{{\tilde{\delta }}_{i}}^{T}{\tilde{\delta }}_{i}+\frac{1}{2\kappa_{i2}}\Xi_{i}+\frac{\eta_{i}}{2\kappa_{i1}}{{\bar{\delta }}_{i}}^{2}+\frac{1}{2\kappa_{i1}}{\bar{\delta }}_{i0}^{2}+\frac{\Upsilon_{i}}{2\kappa_{i2}}\sum_{k=1}^{3}\Gamma_{ik,max}^{2} \end{array}\;\;$$
where $S_{i}=\frac{\beta_{i2}}{2\gamma_{i2}}\Xi_{i2}+\frac{1}{2\gamma_{i2}}{\bar{\Xi }}_{i2}+\frac{\lambda_{iA}}{2}{\bar{D}}_{iA}^{2}+3\kappa_{0}{\bar{D}}_{iA}\mu_{i2}$.

Now, denote ${\bar{S}}_{i}=S_{i}+\frac{1}{2\kappa_{i2}}\Xi_{i}+\frac{\Upsilon_{i}}{2\kappa_{i2}}\sum_{k=1}^{3}\Gamma_{ik,max}^{2}+\frac{\eta_{i}}{2\kappa_{i1}}{\bar{\delta }}_{i}^{2}+\frac{1}{2\kappa_{i1}}{\bar{\delta }}_{i0}^{2}$, $s_{i}=min\left(K_{im}^{*},\beta_{iA}-1,\xi_{iA}\lambda_{iA},2(K_{imE}-1),\Upsilon_{i}-1,\eta_{i}-1\right)$, where m=1,2, $s_{i}>0$. From (57), we have
(58)$${\dot{V}}_{i2}^{*}\leq -s_{i}V_{i2}^{*}+{\bar{S}}_{i}\;\;$$

**Theorem 2.** 
*Under the Assumptions 4–6, with the adaptive estimation laws (46)–(49) and control laws (43), (45), the attitude subsystem (3) of i-th QR subject to input saturation (3), actuator faults (4) and state constraints (5), possesses the following properties:*
i.
*The attitude state constraints (5) of i-th QR will not be exceeded during formation flight.*
ii.
*The attitude and angular velocity tracking error will exponentially converge into the set $∥z_{im}∥\leq \sqrt{\frac{2{\bar{S}}_{i}}{s_{i}}},m=1,2$.*
iii.
*The estimation errors ${\tilde{\Gamma }}_{i},\;{\tilde{\delta }}_{i}$, $\Delta {\tilde{F}}_{i}$, ${\tilde{D}}_{iA}$ and the closed-loop signals $E_{im}$ will be bounded, m=1,2.*



**Proof.** By (58), we have $V_{i2}^{*}\leq (V_{i2}^{*}\left(0\right)-\frac{{\bar{S}}_{i}}{s_{i}})e^{-s_{i}t}+\frac{{\bar{S}}_{i}}{s_{i}}$; thus, $V_{i2}^{*}$ has upper bound, which means the BLF is bounded. Besides,
(59)$${∥z_{im}∥}^{2}\leq \frac{2C_{im}^{2}}{\pi }\tan^{-1}(\frac{\pi }{C_{im}^{2}}(V_{i2}^{*}\left(0\right)-\frac{{\bar{S}}_{i}}{s_{i}})e^{-s_{i}t}+\frac{\pi }{C_{im}^{2}}\frac{{\bar{S}}_{i}}{s_{i}})<\frac{2C_{im}^{2}}{\pi }\frac{\pi }{2}=C_{im}^{2}\;\;$$
then we get $∥z_{im}∥<C_{im}$, m=1,2. Due to $A_{i}=z_{i1}+A_{iC}$ and $\Omega_{i}=z_{2}+\alpha_{i}$, we have $∥A_{i}∥\leq ∥z_{i1}∥+∥A_{iC}∥<{\bar{C}}_{i1}-C_{id}+C_{id}={\bar{C}}_{i1}$ and $∥\Omega_{i}∥\leq ∥z_{i2}∥+∥\alpha_{i}∥<{\bar{C}}_{i2}-{\bar{\alpha }}_{i}+{\bar{\alpha }}_{i}={\bar{C}}_{i2}$; hence, during formation flight, no violation of attitude state constraints will occur. Additionally,
(60)$$\frac{1}{2}z_{im}^{T}z_{im}\leq \frac{C_{im}^{2}}{\pi }\tan(\frac{\pi z_{im}^{T}z_{im}}{2C_{im}^{2}})\leq (V_{i2}^{*}\left(0\right)-\frac{{\bar{S}}_{i}}{s_{i}})e^{-s_{i}t}+\frac{{\bar{S}}_{i}}{s_{i}}\;\;$$
where m=1,2, which indicates that $z_{im}$ will exponentially converge into $∥z_{im}∥\leq \sqrt{\frac{2{\bar{S}}_{i}}{s_{i}}}$, and the estimation errors and closed-signals mentioned above will also be bounded. □

## 4. Simulations

To demonstrate the effectiveness of the proposed scheme, some comparative simulations were carried out, which were programmed via Matlab 2016a and performed on a PC with a 4-core Intel i7-4980HQ@2.8 GHz CPU and 16 GB of RAM. The application scenario of using 5 QRs to enclose and cover a moving ground target is considered. Suppose a target is detected at t=0 s and moving along $T_{g}={\left[15sin\left(0.026t\right),15cos\left(0.026t\right),0\right]}^{T}$. Meanwhile, the QRs will follow the virtual leader to fly right above the target and cover its adjacent area to monitor or sense. Then, the QR formation will converge towards its center at $T_{1}$, start spinning at $T_{2}$ and lower the altitude at $T_{3}$ to enclose the target closely. The target’s adjacent area is defined as a circular area with the radius being 3.5 m and centered on the target. The coverage area of *i*-th QR is centered on $\left[P_{i1},P_{i2},0\right]$, with the radius being $P_{i3}tan\frac{\theta_{S}}{2}$, and $\theta_{S}=50^{\circ }$ represents the angle of view of the sensor payload. The trajectory of the virtual leader is set as $P_{d}=\left(1-e^{-2t}\right)T_{g}+{\left[0,0,\left(5+h_{t}\right)\left(1-e^{-0.3t}\right)\right]}^{T}$, with $h_{t}=\frac{2}{\pi }atan\left(T_{3}-t\left(i\right)\right)-1$. The formation function is designed as $\Lambda_{F}={\left[\Lambda_{1}^{T},\Lambda_{2}^{T},\Lambda_{3}^{T}\right]}^{T}$, where $\Lambda_{i}=A_{t}{\left[cos\left(\omega_{t}\left(t-T_{2}\right)+\frac{(4i+1)\pi }{10}\right),sin\left(\omega_{t}\left(t-T_{2}\right)+\frac{(4i+1)\pi }{10}\right),0\right]}^{T}$, with $A_{t}=\frac{1}{\pi }arctan\left(5\left(-t\left(i\right)+T_{1}\right)\right)+0.5$, $\omega_{t}=\frac{\omega_{0}}{\pi }\left[arctan\left(50\left(t-T_{2}\right)\right)+\frac{\pi }{2}\right]$ and $\omega_{0}=0.8\left(rad/s\right)$. The topology graph is shown in Figure 4, which is undirected and connected, with the weights being $a_{12}=a_{21}=1$, $a_{23}=a_{32}=1$, $a_{34}=a_{43}=1$, $a_{45}=a_{54}=1$ and $b_{1}=b_{5}=1$.

**Remark** **7.**
*In practical applications, the motion information of some non-cooperative targets may not be directly obtained. In this case, the estimated motion information can be obtained by other means and used for formation control, but it is not within the scope of this study. More details can be seen in [49,50]. $\left(\Lambda_{F},P_{d}\right)$ can be designed carefully according to different sensing tasks, sensor performances and quality-of-service policies. The $\left(\Lambda_{F},P_{d}\right)$ chosen in this paper is a basic example to demonstrate the effectiveness of the proposed method.*


With reference to a physical product, the parameters of the QRs are set as $m_{i}=0.856$ (kg), $J_{i}=diag\{0.02351,0.02351,0.04701\}$ and $K_{iD}=diag\{0.003,0.003,0.003\}$. The time-varying delays are set as $\tau_{1}=0.025+0.01sin\left(0.15t\right)$ (s), $\tau_{2}=0.03+0.0125sin\left(0.15t\right)$ (s) and $\tau_{3}=0.035+0.015sin\left(0.15t\right)$ (s). The initial conditions are $P_{1}={\left[0,17.64,0\right]}^{T}$ (m), $P_{2}={\left[-2.5,15.83,0\right]}^{T}$ (m), $P_{3}={\left[-1.56,12.88,0\right]}^{T}$ (m), $P_{4}={\left[1.54,12.86,0\right]}^{T}$ (m), $P_{5}={\left[2.51,15.8,0\right]}^{T}$ (m), $A_{i}=0$ (rad), and $V_{i}=0$ (m/s), $\Omega_{i}=0$ (rad/s). The constraints on attitude state $A_{i}$ is ${\bar{C}}_{i1}=∥A_{iC}∥+0.065$ (rad) and the angular velocity state $\Omega_{i}$ is constrained by ${\bar{C}}_{i2}=\frac{13\pi }{36}$ (rad/s). The control input saturation of attitude controller is set as $U_{iAmax}={\left[0.4,0.4,2\right]}^{T}$ (Nm), $U_{iAmin}={\left[-0.4,-0.4,-2\right]}^{T}$ (Nm). The position controller parameters are $k_{P}=4.9$, $k_{V}=6.85$, $K_{P}=0.0024$, $K_{V}=0.0036$, $K_{W}=0.001725$, $\kappa_{ik}=1824$ (k=1,2,3). The attitude controller parameters are $K_{i1\alpha }=K_{i2\Phi }=3.9$, $\mu_{i1}=\mu_{i2}=0.0012$. The adaptive laws parameters are $\xi_{iA}=8.5$, $\lambda_{iA}=0.022$, $\kappa_{i1}=0.31$, $\eta_{i}=1.52$, $\kappa_{i2}=0.7$, $\Upsilon_{i}=0.85$. The design constant of command filter are $\omega_{iB}=22$, $\xi_{iB}=1.6$. The parameters for RBFNN are $l_{ik}=10$ and the initial weights are set randomly, where k=1,2,3. The lumped uncertainty terms of position subsystem are given as
(61)$$\begin{array}{c} \left\{ \begin{array}{c} F_{1P}={\left[sin\left(0.2P_{11}+V_{11}\right),sin\left(0.1P_{12}+V_{12}\right),sin\left(0.15P_{13}+V_{13}^{2}\right)\right]}^{T}\\ F_{2P}={\left[sin\left(0.15P_{21}+V_{21}\right),0.9sin\left(0.2P_{22}+V_{22}\right),sin\left(0.1P_{23}+V_{23}^{2}\right)\right]}^{T}\\ F_{3P}={\left[sin\left(0.1P_{31}+V_{31}\right),sin\left(0.2P_{32}+V_{32}\right),0.8sin\left(0.1P_{33}+V_{33}^{2}\right)\right]}^{T}\\ F_{4P}={\left[sin\left(0.12P_{41}+V_{41}\right),sin\left(1.6P_{42}+V_{42}\right),sin\left(0.1P_{43}+V_{43}^{2}\right)\right]}^{T}\\ F_{5P}={\left[sin\left(0.18P_{51}+V_{51}\right),sin\left(1.2P_{52}+V_{52}\right),0.7sin\left(0.12P_{53}+V_{53}^{2}\right)\right]}^{T} \end{array}\right. \end{array}\;\;$$

Besides, the external disturbances of the attitude subsystem containing stable, periodic and aperiodic components are set as follows
(62)$$\begin{array}{c} \left\{ \begin{array}{c} ∥D_{1A}∥=0.015\left(sin\left(t+3.4\right)+cos\left(e^{0.1t}t+6.2\right)\right)+0.01cos\left(5t+1.2\right)+0.18\\ ∥D_{2A}∥=0.015\left(cos\left(t+5.7\right)+cos\left(e^{0.1t}t+5.9\right)\right)+0.01cos\left(5t+4.3\right)+0.2\\ ∥D_{3A}∥=0.015\left(sin\left(t-2.4\right)+cos\left(e^{0.1t}t+4.2\right)\right)+0.01cos\left(5t+0.6\right)+0.22\\ ∥D_{4A}∥=0.015\left(cos\left(t-1.6\right)+cos\left(e^{0.1t}t+5.1\right)\right)+0.01cos\left(5t+0.9\right)+0.26\\ ∥D_{5A}∥=0.015\left(sin\left(t+1.8\right)+cos\left(e^{0.1t}t+3.3\right)\right)+0.01cos\left(5t+1.7\right)+0.24 \end{array}\right. \end{array}\;\;$$

The time-varying multiplicative and additive actuator fault signals are considered as follows
(63)$$\left\{ \begin{array}{c} \Gamma_{1}=diag\{0.7,0.85,0.77\}+diag\{0.3,0.15,0.23\}e^{-t}\\ \Gamma_{2}=diag\{0.8,0.75,0.85\}+diag\{0.2,0.25,0.15\}e^{-t}\\ \Gamma_{3}=diag\{0.69,0.75,0.82\}+diag\{0.31,0.25,0.18\}e^{-t}\\ \Gamma_{4}=diag\{0.73,0.8,0.81\}+diag\{0.27,0.22,0.19\}e^{-t}\\ \Gamma_{5}=diag\{0.82,0.78,0.86\}+diag\{0.18,0.22,0.14\}e^{-t} \end{array}\;\;\right.$$
(64)$$\left\{\begin{array}{c} \delta_{1}={\left[0.1,0.1,-1.3\right]}^{T}\left(1-e^{-t}\right)+{\left[0,0.05sin\left(0.27t\right),0.5\right]}^{T}\\ \delta_{2}={\left[0.15,0.1,1\right]}^{T}\left(1-e^{-t}\right)+{\left[0,0.05cos\left(0.27t\right)\left(1-e^{-0.05t}\right),-0.5\right]}^{T}\\ \delta_{3}={\left[-0.1,0.1,-1.2e^{-0.05t}\right]}^{T}\left(1-e^{-t}\right)+{\left[0,0.05sin\left(0.27t\right)e^{-0.03t},0.7e^{-0.05t}\right]}^{T}\\ \delta_{4}={\left[-0.15,0.1,0.8\right]}^{T}\left(1-e^{-t}\right)+{\left[0,0.055sin\left(0.2t\right)\left(1-e^{-0.05t}\right),-0.6\right]}^{T}\\ \delta_{5}={\left[0.12,0.12,0.7\right]}^{T}\left(1-e^{-t}\right)+{\left[0,0.045cos\left(0.15t\right)\left(1-e^{-0.07t}\right),-0.4\right]}^{T} \end{array}\right.\;\;$$
with ${\hat{\Gamma }}_{ik}=1$, ${\hat{\delta }}_{ik}=0$, i,k=1,2,3 as the initial estimation values.

The simulation results of trajectory, position, attitude, attitude constraints, angular velocity constraints, control inputs, RBFNN, disturbance estimation, multiplicative fault estimation and additive fault estimation are demonstrated in Figure 5, Figure 6, Figure 7, Figure 8, Figure 9, Figure 10, Figure 11, Figure 12, Figure 13 and Figure 14, respectively. The trajectory and position snapshots of QRs and a moving target are illustrated in Figure 5. It can be seen that the QRs can successfully form the desired formation pattern $\Lambda_{F}$ and track the desired trajectory $P_{d}$, thereby achieving the full coverage and close-range enclosing. Figure 6 shows the position tracking errors with and without RBFNN. In the case of with RBFNN, the tracking errors converge to the neighborhood of zero rapidly under the influence of lumped uncertainties. The effectiveness of RBFNN is demonstrated by the fact that tracking error cannot be reduced to near zero and continues to oscillate in the absence of RBFNN. Figure 7 demonstrates the robust learning ability of RBFNNs, convergence of approximation errors takes only a few seconds and oscillation at the beginning is caused by randomly selected initial weights. Figure 8 depicts the tracking performance of AFTAC, which, despite initial misalignments, tracks the command signal exceptionally well. Furthermore, Figure 9 and Figure 10 show the norm of attitude $∥A_{i}∥$ and norm of angular velocity $∥\Omega_{i}∥$ always satisfy the predefined constraints ${\bar{C}}_{i1}$ and ${\bar{C}}_{i2}$ during the whole process. In Figure 9, the unconstrained AFTAC in [51] is compared under identical conditions, and the parameters of the comparison AFTAC are adjusted to achieve relatively good tracking performance. One can observe that the comparison AFTAC tracks the command signal closely throughout the whole process, but it cannot guarantee the state constraints will always be met; the constraints are occasionally exceeded, particularly when the command signal changes rapidly. The comparison results demonstrate that the specific system states can be constrained within a certain range to meet safety or sensor payload requirements, which is an advantage of our method. Figure 11 depicts the input signals of QRs, which contain large spikes at the beginning, $T_{1}$ and $T_{2}$. These spikes are effectively filtered out by input saturation, where the actuator’s limitations are fully reflected. As demonstrated by the proof of Theorem 2, the upper bound of external disturbance and actuator fault signals are effectively estimated in Figure 12, Figure 13 and Figure 14.

## 5. Conclusions

This article presents a distributed formation control scheme for a group of QRs subject to constraints and time-varying delays. The proposed scheme consists of NTDPC for position control and state-constrained AFTAC for attitude regulating. In NTDPC, an adaptive RBFNN is utilized to compensate the lumped uncertainties, and a Lyapunov–Krasovskii analysis is applied to handle the time-varying delay. Based on the backstepping technique, AFTAC employs a tan-type BLF to handle the state constraints, an auxiliary system combined with a command filter to deal with input saturation and adaptive estimators to compensate fault signals and disturbances. To determine the efficacy of the proposed method, comparative simulations were conducted. We demonstrate that the proposed method can be applied for a mobile sensing task; the formation tracking errors are UUB; the estimation errors of actuator faults, uncertainties, and disturbances are also bounded; and the predefined constraints will never be violated during formation flight. However, the current method has some limitations, such as symmetric state constraints and a fixed network topology. Additional research will yield asymmetric state constraints and a mechanism for switching topologies.

## Figures and Tables

**Figure 1 sensors-22-07497-f001:**
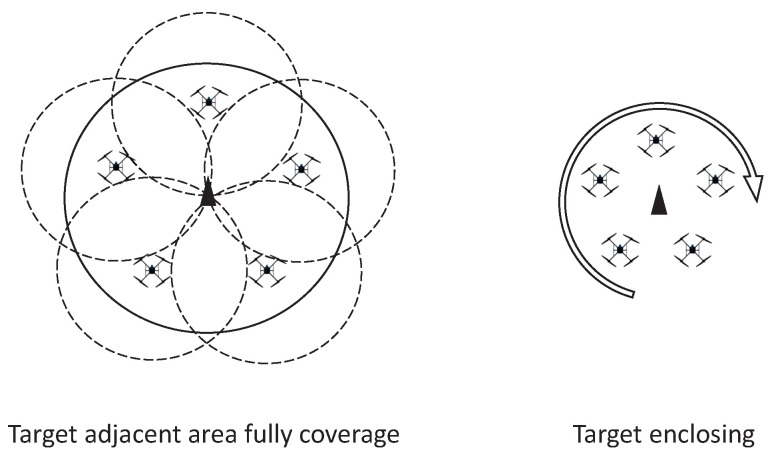
Depiction of the target-enclosing and covering task.

**Figure 2 sensors-22-07497-f002:**
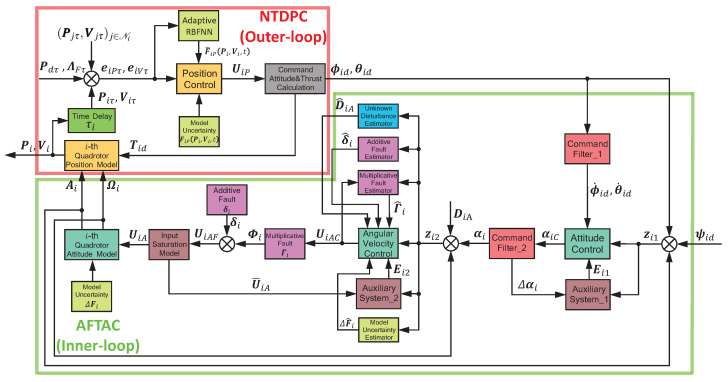
Block diagram of proposed formation control scheme scheme.

**Figure 3 sensors-22-07497-f003:**
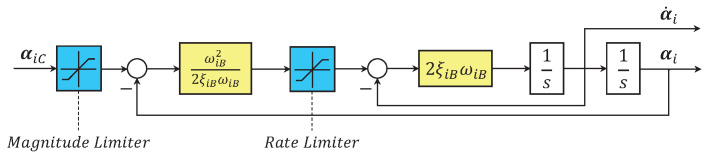
Framework of the command filter, with $\omega_{iB}$ and $\xi_{iB}$ being design constants, i∈Σ.

**Figure 4 sensors-22-07497-f004:**
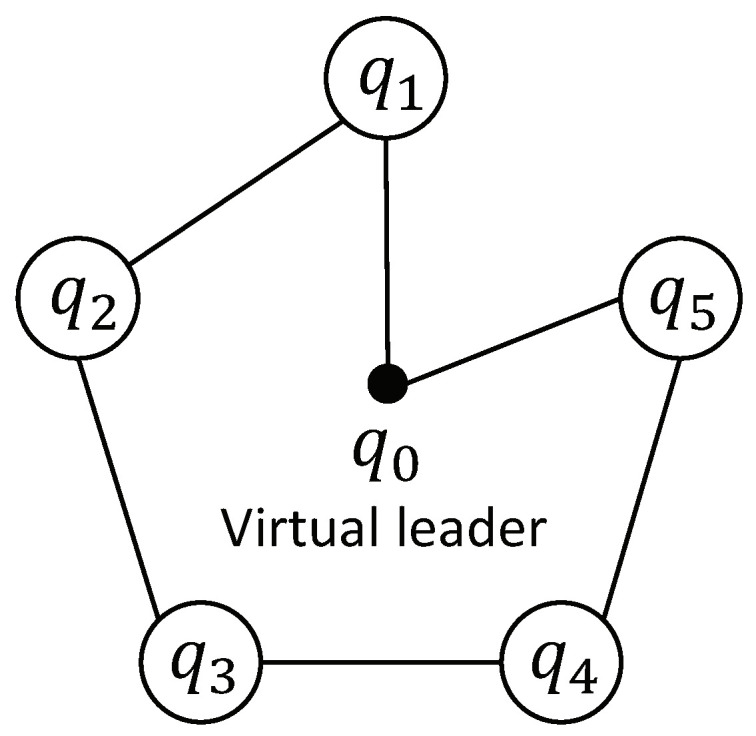
The communication topology graph.

**Figure 5 sensors-22-07497-f005:**
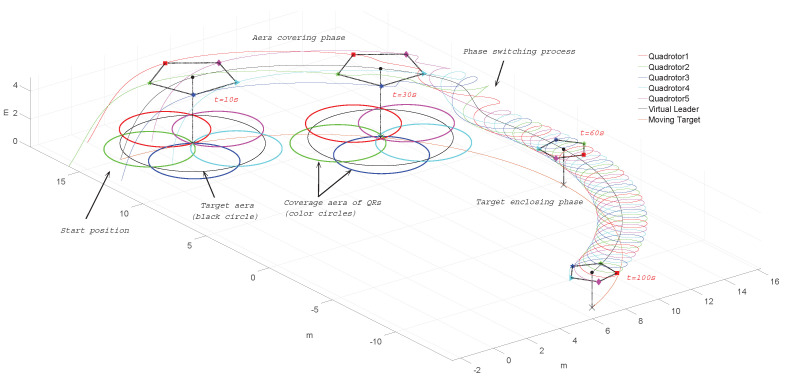
Trajectory, position snapshots and coverage areas of Quadrotors (QRs) and a moving target with $T_{1}=35s$, $T_{2}=40s,T_{3}=45s$.

**Figure 6 sensors-22-07497-f006:**
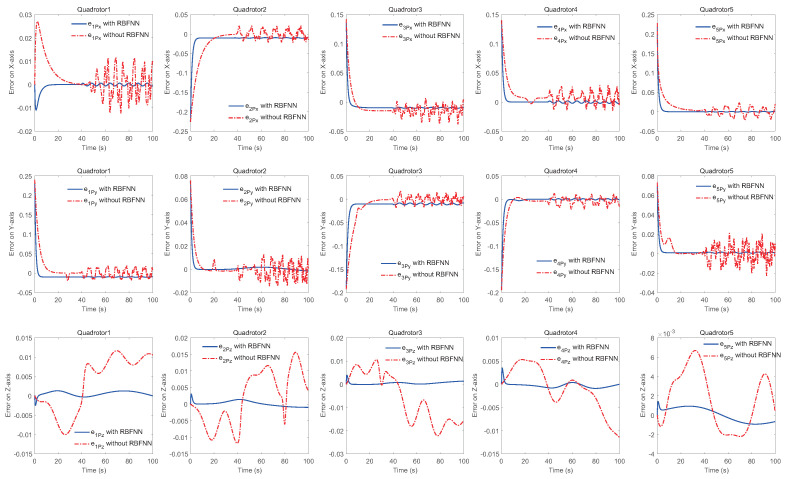
Comparison of position tracking errors with and without radial basis function neural network (RBFNN).

**Figure 7 sensors-22-07497-f007:**
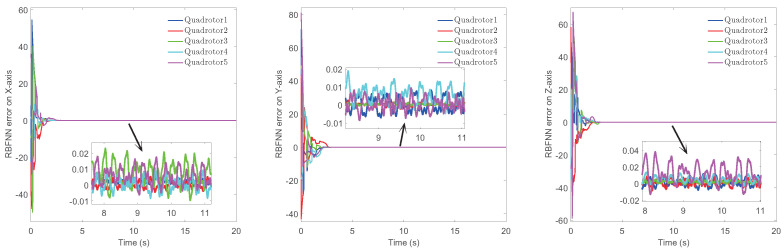
RBFNN approximation errors on 3 axes.

**Figure 8 sensors-22-07497-f008:**
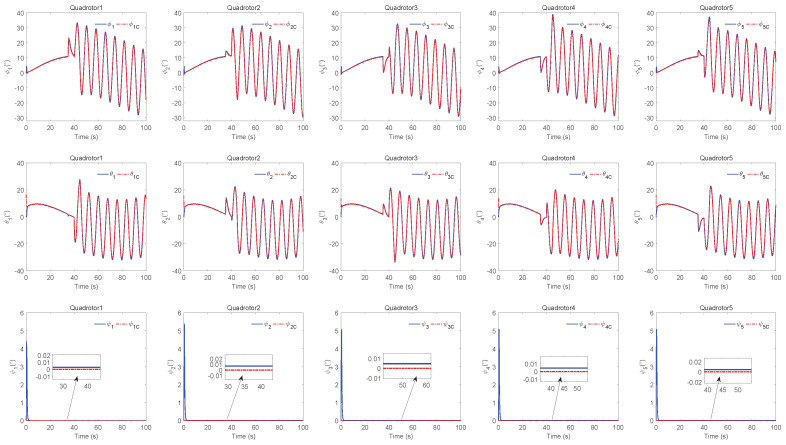
Attitude signals of QRs.

**Figure 9 sensors-22-07497-f009:**
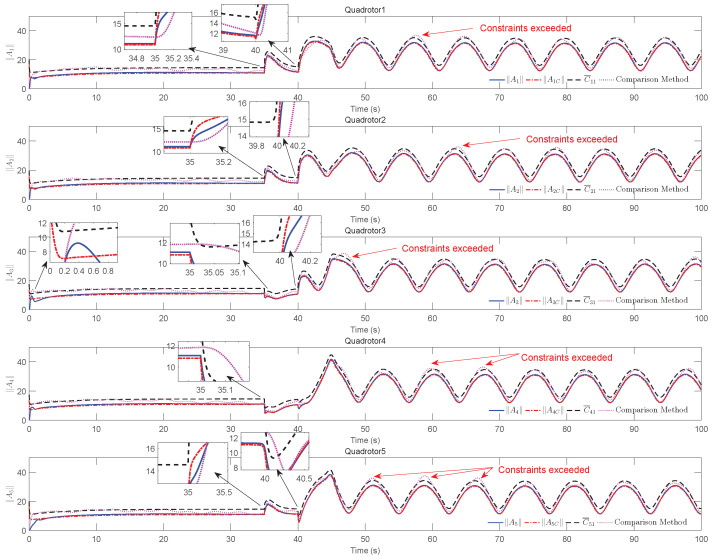
$∥A_{i}∥$, $∥A_{iC}∥$ and constraints ${\bar{C}}_{i1}$, i=1,2,3,4,5.

**Figure 10 sensors-22-07497-f010:**
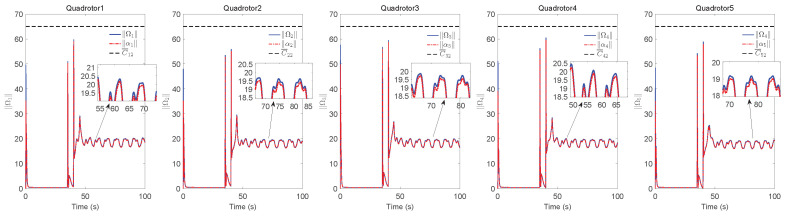
$∥\Omega_{i}∥$, $∥\alpha_{i}∥$ and constraints ${\bar{C}}_{i2}$, i=1,2,3,4,5.

**Figure 11 sensors-22-07497-f011:**
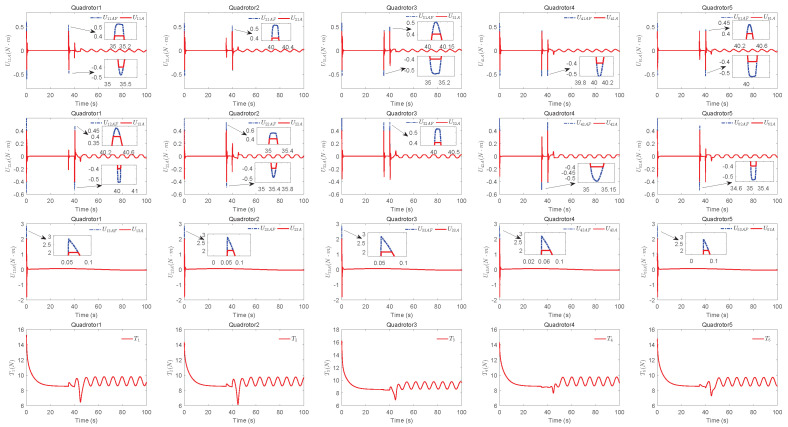
Input signals of QRs.

**Figure 12 sensors-22-07497-f012:**
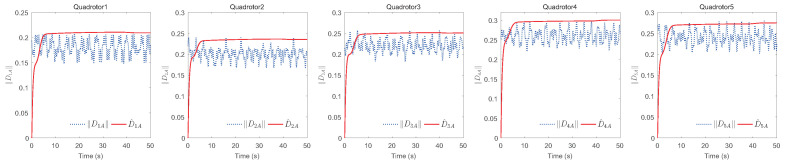
Estimation of external disturbances’ upper bounds.

**Figure 13 sensors-22-07497-f013:**
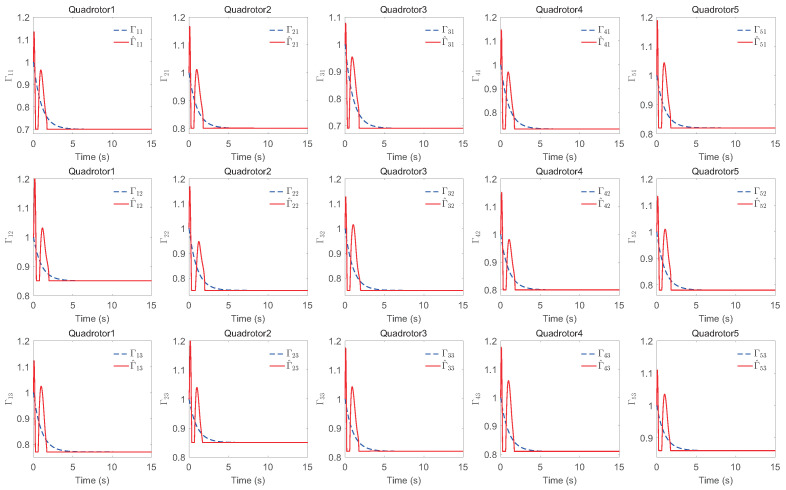
Estimation of multiplicative faults.

**Figure 14 sensors-22-07497-f014:**
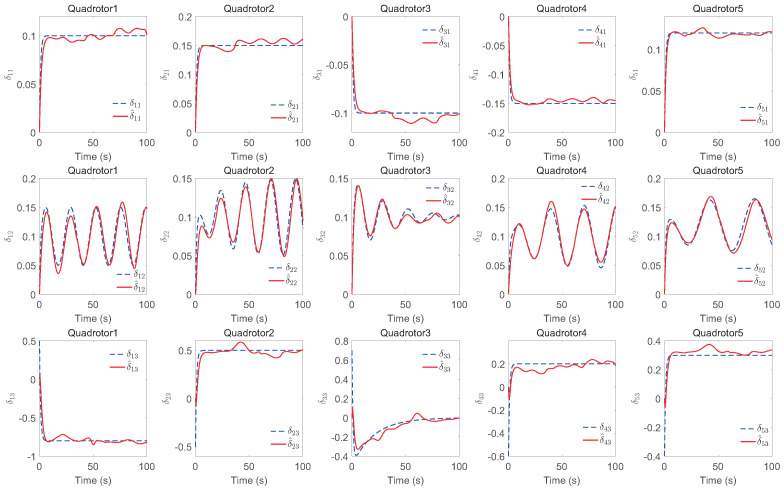
Estimation of additive faults.

## Data Availability

The data that support the findings of this study are available from the corresponding author upon reasonable request.

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
