# Peer review of "Target Enclosing and Coverage Control for Quadrotors with Constraints and Time-Varying Delays: A Neural Adaptive Fault-Tolerant Formation Control Approach"

_sensors, 2022, doi:10.3390/s22197497_

Round 1

Reviewer 1 Report

This is a simulation/model study of the formation fault tolerant control for multiple quadrotors. The focus is on the input saturation, state constraints and time-varying delays. The writing needs improvement.  The figures are not clear and must be improved. The study and results in the manuscript are not very related to sensors. It is more appropriate for other journals related to controls and robotics.

Author Response

First of all, I thank you for your useful commets.

1.In the revised wertion, the writing will be further improved.

2.The Y-axis of figures will be increased to show more details.

3.In fact, UAV formation has a very close relationship with sensor applications. Multi UAV system is a powerful platform for mobile sensing applications. More  sensor related content will be added to the revised paper, and new simulation will be in combination with the actual application scenarios of mobile sensing, such as coverage control and target encirclement control.

Reviewer 2 Report

The manuscript is very well prepared, with detailed explanation and analysis of the proposed method.

The main weakness of the manuscript is that the proposed method has been verified by only one simulation scenario which is also simple (3 QR with smooth and harmonical trajectories).

Some other minor remarks:

- Line 87: wij is not defined.

- Line 88: would it be "aij = aji = ∞" instead of 0?

- A reference should be added for the QR model in Eqs (1) and (2).

- In Eq (18), would the * be 0?

Author Response

First of all, thank you for your careful and valuable comments!

1.Indeed, the simulation in my original draft only used three UAVs. My original idea was that the number of UAVs in my algorithm did not affect its theoretical stability, so only three UAVs were used for simplicity. However, in order to further demonstrate the effectiveness of the proposed method, I still adopted your suggestions in the revised version, increasing the number of UAVs from three to five. Your suggestions are very useful!

2.As for the problem that the trajectory you mentioned is too harmonious and simple, I think you may not have noticed that in the original simulation, at T0, the formation of UAVs suddenly began to rotate around the virtual pilot. You can see in the control input curve that moment, the input saturation phenomenon appeared, and the attitude angular velocity of UAVs approached the preset constraints. This is what I specifically set up to generate an disharmonious transient process, It is intended to illustrate the effectiveness of state constrained control and anti saturation measures. And the UAV will not always be in the limit state in the actual application process, so the trajectory I choosed was not always in the limit state. But based on your valuable suggestions, in the new simulation, I added one more inharmonious transient process, which can further illustrate the problem.

3. Line 87, "wij" was a mistake in writing, it should be "aij", which represents the element of weighted adjacency matrix,and has been corrected.

4. Line 88, "aij = aji = 0", this is correct. “aij” represents the degree of association between node "i" and node "j". The closer the two nodes are associated, the greater the "aij" would be. Therefore, when two nodes have no communication relationship, their corresponding "aij" would be zero.

5. Relevant references have been added for models in Eqs (1) and (2) in revised vertion.

6. Yes, the "*" in Eq(18) represents for "0", the asterisk is used to indicate that "M" in Eq(18) is a symmetric matrix, but I will change the asterisk to "0" to avoid misunderstanding.

And again, thank you for your comments.

Reviewer 3 Report

I have some observations and questions in the attached document.

Author Response

Thank you very much for your detailed comments. Please see the attachment.

Round 2

Reviewer 1 Report

the issues were addressed in the revision,

Reviewer 3 Report

In my opinion, this version of the article Target enclosing and coverage control for Quadrotors with Constraints and Time-Varying Delays: A Neural Adaptive Fault Tolerant Formation Control Approach ,is clearer and has improved. Thank you very much.